# Physical deep learning with biologically inspired training method: gradient-free approach for physical hardware

Mitsumasa Nakajima ®[1] ✉, Katsuma Inoue ®[2] ✉, Kenji Tanaka ®[1],
Yasuo Kuniyoshi[2,3], Toshikazu Hashimoto[1] & Kohei Nakajima ®[2,3] ✉

Ever-growing demand for artificial intelligence has motivated research on unconventional computation based on physical devices. While such computation devices mimic brain-inspired analog information processing, the learning procedures still rely on methods optimized for digital processing such as backpropagation, which is not suitable for physical implementation. Here, we present physical deep learning by extending a biologically inspired training algorithm called direct feedback alignment. Unlike the original algorithm, the proposed method is based on random projection with alternative nonlinear activation. Thus, we can train a physical neural network without knowledge about the physical system and its gradient. In addition, we can emulate the computation for this training on scalable physical hardware. We demonstrate the proof-of-concept using an optoelectronic recurrent neural network called deep reservoir computer. We confirmed the potential for accelerated computation with competitive performance on benchmarks. Our results provide practical solutions for the training and acceleration of neuro-morphic computation.

Machine learning based on artificial neural networks (ANNs) has successfully demonstrated its excellent ability through record-breaking performance in image processing, speech recognition, game playing, and so on[1–3]. Although these algorithms resemble the workings of the human brain, they are basically implemented on a software level using conventional von Neumann computing hardware. However, such digital-computing-based ANNs are facing issues regarding energy consumption and processing speed[4]. These issues have motivated the implementation of ANNs using alternative physical platforms[5], such as spintronic[6–8], ferroelectric[9,10], soft-body[11,12], photonic hardware[13–18], and so on[19–22]. Interestingly, even passive physical dynamics can be used as a computational resource in randomly connected ANNs. This framework is called a physical reservoir computer (RC)[21–23] or an extreme learning machine (ELM)[24–26], whose ease of implementation has greatly expanded the choice of implementable materials and its application range. Such physically implemented neural networks (PNNs) enable the outsourcing of the computational load for specific tasks to a physical system such as a memory[27], optical link[28,29], sensor component[30,31], or robotic body[32]. The experimental demonstrations of these unconventional computations have revealed performance competitive with that of conventional electronic computing[33–35].

Constructing deeper physical networks is one promising direction for further performance improvement because they can extend network expression ability exponentially[36,37], as opposed to the polynomial relationship in wide (large-node-count) networks. This has motivated proposals of deep PNNs using various physical platforms[14,16,30,38–43]. Their training has basically relied on a method called backpropagation (BP), which has seen great success in the software-based ANN.

[1]NTT Device Technology Labs., 3-1 Morinosato-Wakamiya, Atsugi, Kanagawa 243-0198, Japan. [2]Graduate School of Information Science and Technology, The University of Tokyo, 7-3-1 Hongo, Bunkyo-ku, Tokyo 113-8656, Japan. [3]Next Generation Artificial Intelligence Research Center, The University of Tokyo, 7-3-1 Hongo, Bunkyo-ku, Tokyo 113-8656, Japan. ✉e-mail: mitsumasa.nakajima.wc@hco.ntt.co.jp; k-inoue@isi.imi.i.u-tokyo.ac.jp; k-nakajima@isi.imi.i.u-tokyo.ac.jp

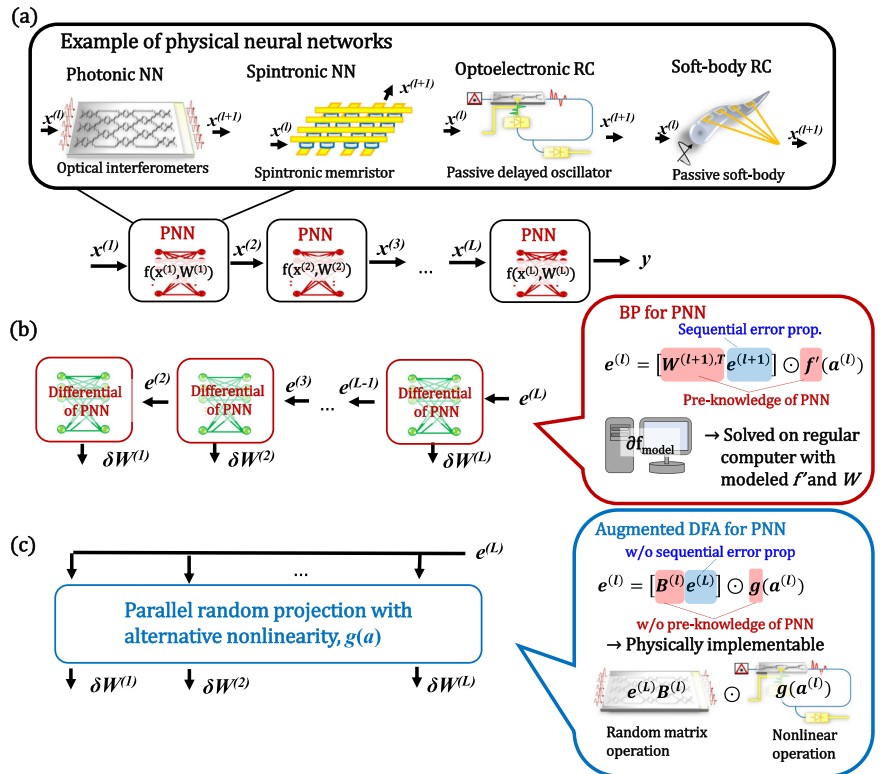

**Fig. 1 | Concept of PNN and its training by BP and augmented DFA. a** Schematics of physical neural networks (PNNs). Training sequence of PNN with **b** BP, and **c** augmented biologically plausible training called direct feedback alignment (DFA). Augmented DFA enables parallel, scalable, and physically accelerable training of deep physical networks based on random projection with alternative non-linearity $g(a)$.

However, BP is not suitable for PNNs in the following respects. First, the physical implementations of the BP operation are still complex and unscalable[40–43]. Thus, the calculation for BP for a PNN is typically executed on an external regular computer with a simulation model of a physical system[14,16,30,39,44]. This strategy results in a loss of any advantage in speed or energy associated with using the physical circuit in the training process. Thus, this method is not suitable for in-situ (online) training; it is only usable for "train once and infer many times" application. Second, BP requires accurate knowledge about the whole physical system. Thus, the performance of the PNNs entirely relies on the model representation or measurement accuracy of the physical system[45]. In addition, when we apply BP to RC, these requirements spoil the unique features of physical RC, i.e. we need to know and simulate a black-boxed physical random network accurately.

Like BP in PNNs, the operational difficulty of BP in biological neural networks has also been pointed out in the brain science field; the plausibility of BP in the brain—the most successful analog physical computer—has been doubted[46–48]. These considerations have motivated the development of biologically plausible training algorithms[49–52]. One promising recent direction is direct feedback alignment (DFA)[53–55]. In this algorithm, fixed random linear transformations of the error signal at the final output layer are employed instead of the backward error signals. Thus, this approach does not require layer-by-layer propagation of error signals or knowledge of the weight. In addition, it has been reported that DFA scales to modern large-scale network models[54]. The success of such biologically motivated training suggests that there is a more suitable way than BP to train PNNs. However, DFA still requires the derivative $f'(a)$ of a nonlinear function $f(x)$ for the training, which hinders the application of DFA methods to physical systems. Although previous studies on DFA training for spiking neural networks (SNNs) have reported that an approximated function can be used as an alternative[56], this approach still requires modeling and simulation of

the physical system. Thus, a more drastic extension of DFA is important for PNN applications.

In this paper, we demonstrate physical deep learning by augmenting the DFA algorithm. In the augmented DFA, we replace the differential of physical nonlinear activation $f'(a)$ in the standard DFA with arbitrary nonlinearity $g(a)$ and show that the performance is robust to the choice of $g(a)$. Owing to this augmentation, we no longer need to simulate $f'(a)$ accurately. As the proposed method is based on parallel random projection with arbitrary nonlinear activation, we can execute the computation for the training on a physical system in the same manner as with the physical ELM or RC concept[21–23]. This enables the physical acceleration of both inference and training. To demonstrate the proof-of-concept, we constructed an FPGA-assisted optoelectronic deep physical RC as a benchtop. Although our benchtop is simple and easy to apply to various physical platforms with only software-level updates, we achieved performance comparable to that of large-scale complex state-of-the-art systems. Moreover, we compared the whole processing time, including that for digital processing, and found the possibility of physical acceleration of the training procedure. We also numerically found that the proposed augmented DFA is applicable to various network models, including more practical architecture and SNNs, suggesting the scalability of our approach. Our approach provides a practical alternative solution for the training and acceleration of neuromorphic physical computation.

## Results
### Direct feedback alignment and its augmentation for physical deep learning
Fig. 1a shows the basic concept of PNNs. The forward propagation of a standard multilayer network is described as $x^{(l+1)} = f(a^{(l)})$, where $a^{(l)} = W^{(l)}x^{(l)}$ with the weight $W^{(l)} \in \mathbb{R}^{N^{(l+1)} \times N^{(l)}}$ and input $x^{(l)} \in \mathbb{R}^{N^{(l)}}$ for the $l$th layer, and $f$ denotes an element-wise nonlinear activation. In the PNN framework, this operation is executed on a physical system;

i.e. $x^{(l)}$, $W^{(l)}$, and $f$ correspond to the physical inputs (e.g., optical intensity, electric voltage, vibration), physical interconnections (e.g., optical, electrical, or mechanical coupling) in the physical system, and physical nonlinearity (e.g., nonlinear optical/magnetic/mechanical effects), respectively. To train such networks, we need to update $W^{(l)}$ to reduce given cost function $E$. A general solution is the BP algorithm shown in Fig. 1b. The gradients for BP are obtained through the chain-rule as follows:

$$e^{(l)} = \left[ W^{(l+1),T} e^{(l+1)} \right] \odot f'(a^{(l)}), \qquad (1)$$

where $e^{(l)} \epsilon \mathbb{R}^{N^{(l+1)}}$ is the error signal at the $l$th layer, defined as $e^{(l)} = \partial E / \partial a^{(l)}$ with $a^{(l)} = W^{(l)} x^{(l)}$, the superscript $T$ denotes transposition, and $\odot$ denotes the Hadamard product. From Eq. (1), we can compute the gradient for each $W^{(l)}$ as $\delta W^{(l)} = -e^{(l)} x^{(l),T}$. The training using Eq. (1) is typically executed on a regular external computer by constructing a physical simulation model[14,16,30,39,44], which incurs large computational cost. Thus, this strategy is not suitable for in-situ training. In addition, the error in the simulation model significantly affects PNN performance. Therefore, the training method for PNNs is still under consideration despite the success of BP in software-based ANNs.

Let us consider DFA as an alternative solution [see Fig. 1c]. In the standard DFA framework, the weighted gradient signals in Eq. (1) are replaced with a linear random projection of the error signal at the final layer $L$[53,54]. Then, we can obtain the following update rule:

$$e^{(l)} = \left[ B^{(l)} e^{(L)} \right] \odot f'(a^{(l)}), \qquad (2)$$

where $B^{(l)} \epsilon \mathbb{R}^{N^{(l+1)} \times N^{(y)}}$ is a random projection matrix for the $l$th layer update, and $f'$ denotes the gradient of $f$. As shown in Eq. (2), we can estimate the gradient without information about $W^{(l)}$. In addition, physical implementation of random projection process $B^{(l)} e^{(L)}$ can be implemented by using various devices because this process is the same as in the physical ELM and RC approach. By using commercially available photonic components, we can emulate $5 \times 10^5$ by $5 \times 10^5$ matrix operations on a single integrated optics[57], which are enough for the single hidden layer even in the state-of-the-art model. Note that in this hypothesis, we assumed a liquid-crystal-on-silicon (LCOS) spatial-light modulator (SLM) for the input encoding and a passive optical diffuser for the random matrix. Thus, the input vector ($e^{(L)}$ for the DFA processing) is reconfigurable, but the random matrix ($B^{(l)}$ for the DFA processing) is not. By using an additional SLM instead of the random scattering medium, we can implement programmable random matrices[34,58]. The limitations due to such a photonic implementation are discussed in Supplementary Information S.9. Also, the photonic acceleration of this process has already been demonstrated[59]. In addition, the DFA process can be parallelized because it is not a sequential equation unlike the one for BP. Despite its simplicity, DFA can scale modern neural network models (see Supplementary Information S1 and ref. [54]). However, $f'(a)$ remains in Eq. (2), requiring accurate modeling and simulations, which is the bottleneck in the learning process for PNNs.

Here, we replace the $f'(a)$ function with the function $g(a)$ to investigate the robustness against the choice of $g(a)$. Then, we derive the update rule as

$$e^{(l)} = \left[ B^{(l)} e^{(L)} \right] \odot g(a^{(l)}), \qquad (3)$$

As $f'(a)$ is replaced with $g(a)$, the equation no longer includes the knowledge for the parameters in the forward propagation. The gradient $\delta W^{(l)}$ can be estimated from the final error $e^{(l)}$ and alternative nonlinear projection of given $a^{(l)}$. Thus, we no longer require the gradient of the original networks, which is highly advantageous for PNN training. As shown in the following section, we can select a broader

range of $g(a)$. The only requirement is to avoid the function uncorrelated to $f'(a)$. Notably, the computation of $g(a)$ can also be implementable to a physical system. A concrete example is shown in the Physical Implementation section below. We named this algorithm as *augmented DFA*.

Interestingly, the augmented DFA is also useful for black-box fixed physical networks such as a physical RC, where black-box means that we do not know (or only have rough information about) $W^{(l)}$ and $f$. When we apply the BP algorithm to physical RC, we need to simulate the gradient of the physical system using a regular computer. Thus, we need to open the black-box (need to measure and approximate $W^{(l)}$ and $f$) to estimate the gradients, which spoils the advantage of such a randomly fixed physical network. On the other hand, the augmented DFA can train a physical RC without the BP and knowledge about the physical system. Although the update rule of the augmented DFA for the RC requires additional computation compared with Eq. (3) [see Eqs. (6-11) in Methods], this can be executed on physical hardware in the same manner as forward propagation in an RC. Thus, we can improve the performance of RC while maintaining its unique features. The detailed update rule for the RC is described in Methods, and the concrete experimental demonstration is shown in the following section.

For the actual physical implementation using the augmented DFA, it should be mentioned how $a^{(l)}$ is obtained from the PNNs for the computation of Eq. (3). For most feedforward-type PNNs, the physical (or electrical) nonlinear layer and linear multiply-accumulate layer (e.g., the fully connected layer) are separated. Thus, we can measure $a^{(l)}$ as an output of a physical network and we can use it as an input parameter for Eq. (3). On the other hand, some physical networks cannot separate the nonlinear and linear layer. For example, an RC includes nonlinearity in the physical dynamics itself. In this case, we can directly obtain $g(a^{(l)})$ by changing the nonlinearity in the physical system. A concrete example of the former (separated physical nonlinearity) and latter (physical nonlinearity included) cases are shown in Supplementary Information S2 and the Physical Implementation section below.

## Basic characterization of augmented DFA

First, we investigated the effect of the augmentation of DFA, that is, the effect of $g(a)$. For this purpose, we used the standard image classification benchmark called the Modified National Institute of Standards and Technology database (MNIST) task with a simple multilayer-perceptron (MLP) model. In the experiment, the MLP model was composed of four fully connected layers with 800 nodes for each layer and two types of nonlinear activation $f(a)$, namely a hyperbolic tangent (tanh), sine (sin), and cosine (cos) function. These nonlinearities correspond to a simple model of common photonic implementation. In this experiment, we generated $g(a)$ from some well-used functions (sine, cos, tanh, triangle) and the random Fourier series $g(a) = \sum_{k=1}^{K} p_k \sin(ka\pi) + q_k \cos(ka\pi)$, where $p_k$ and $q_k$, are the random uniform coefficients sampled from $\mathbb{R} \in [-1:1]$. $K$ was set to 4 and normalized by the relationship $\sum_{k=1}^{K} |p_k| + |q_k| = 1$. A hundred random Fourier series were examined in this experiment. Random matrix $B^{(l)}$ was generated from the uniform distribution.

The training curves for the augmented DFA and BP are shown in Fig. 2a. For comparison, we also show the results for the BP algorithm in Eq. (1) with $f'(a)$ replaced with $g(a)$. Here, the $g(a)$ was generated from the various nonlinear activations and the random Fourier series with 100 random seeds. Thus, correlation coefficient (corr) $\eta$ between $f'(a)$ and $g(a)$ varied with the difference in the random seed and the choice of nonlinearity (see [Eq. (12)] in Methods for the definition). The solid line and the shaded area indicate the averaged test error for all the experiments and the region between the maximum and minimum values, respectively. The color difference indicates the difference in the examined nonlinearity in the forward propagation $f$. As can be

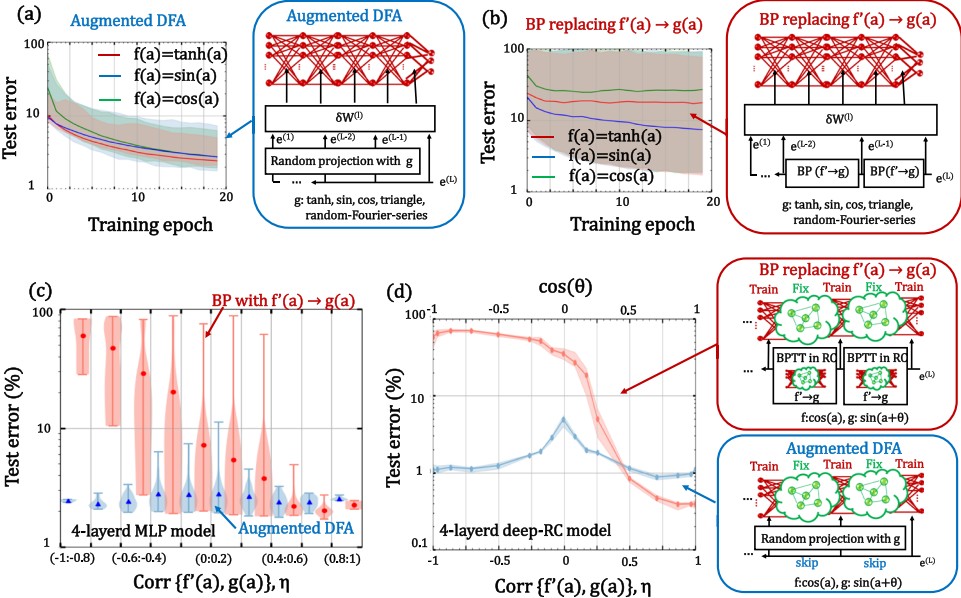

**Fig. 2 | BP vs augmented DFA.** Training curve of 4-layer MLP with 800 hidden nodes for **a** augmented DFA and **b** BP with various $g(a)$:$\sin(a)$, $\cos(a)$, triangle$(a)$, $\tanh(a)$ functions, and 40 random Fourier series. Color differences indicate differences in activation function [$f(a)$=$\sin(a)$, $\cos(a)$, $\tanh(a)$]. Bold line and shaded area indicate the average and maximum-minimum region, respectively. **c** Test error distribution of four-layer fully connected neural network as a function of the correlation coefficients between $f'(a)$ and $g(a)$, $\eta$. Blue and red boxplots in **c** are the results for the model trained by BP and augmented DFA, respectively. $\eta$ was scanned by using various $g(a)$:$\sin(a)$, $\cos(a)$, triangle$(a)$, $\tanh(a)$ functions and a random Fourier series. The whiskers, plots, and filled area show the minimum and

maximum values, average values, and density of data distribution, respectively. **d** Test error of four-layer RC with 400 nodes as a function of $\eta$. The spectral radius of reservoir weight was set to 0.9 because the performance of the deep RC was maximized in this region (see Supplementary Information S13). Red and blue plots in **d** are the results for the model trained by BP and augmented DFA. $\eta$ was scanned by using $g(a) = \sin(a+\theta)$ with $\theta = 0$, 15, 30, 45, 60, 75, 80, 85, 90, 95, 100, 105, 120, 135, 150, 165, 180°. As a reference, $\cos(\theta)$ is displayed as a second $x$ axis. Data in this figure were obtained using standard CPU/GPU computation. Each experiment was repeated five times.

seen, the average accuracy of the augmented DFA is far superior to that for BP when we replace the nonlinearity. The training curve of the augmented DFA seems to converge for the examined case, while the one for the BP seems to often diverge. Figure 2c shows the test error as a function of $\eta$. The whiskers, plots, and filled area show the minimum and maximum values, average values, and density of data distribution, respectively. The case for $\eta = 1$ means that $g(a)$ equals $f'(a)$, which corresponds to the case of the standard BP and DFA in Eqs. (1) and (2). The cases for $\eta = 0$ and $-1$ correspond to uncorrelation and negative correlation, respectively. The test error of BP increases sharply when $\eta$ deviates from 1. In particular, almost no meaningful training was possible when the correlation coefficient was negative. This is because the update direction became opposite from the correct direction. On the other hand, the test accuracy for the augmented DFA showed a gentle dependency on $\eta$. These results indicate that the training is highly robust to the choice of $g(a)$. The test error for the augmented DFA was maximized at $\eta = 0$ [i.e., $g(a)$ became the uncorrelated function of $f'(a)$]. Even in the worst case, we were able to obtain an accuracy of about 89%, which is far superior to that for BP. In addition, learning can be performed when the correlation coefficient is negative. We think that this is due to the random weight matrix $B$; that is, the randomly distributed linear projection term in Eq. (3) erases the positive/negative sign of $g(a)$. Note that the observed convergence of the augmented DFA is not always guaranteed: e.g., we observed divergence of the training curve when $f'(a)$ was a random Fourier series (see Supplementary Information S5). However, the robustness against the choice of $g(a)$ was much superior to that for BP in all examined cases.

One index for evaluating whether the feedback alignment algorithm operates well or not is the alignment angle ($\angle\delta_{BP/DFA}$), which is the angle between $\delta_{BP}$ and $\delta_{DFA}$, where $\delta_{BP}$, $\delta_{DFA}$, and $\angle\delta_{BP/DFA}$ are defined as $\delta_{BP} = W^T e^{(l)}$ and $\delta_{DFA} = B^{(l)} e^{(L)}$, $\angle\delta_{BP/DFA}=\cos^{-1}(\delta_{BP}\cdot\delta_{DFA})$[S2]. When the alignment angle lies within 90°, the network trained by the

augmented DFA is roughly in the same direction as the BP would be. Here, we analyzed the alignment angle of the network with four hidden layers (see Supplementary Information S11 for details). We found that the alignment angles for the BP are significantly increased when $\eta$ is apart from one, which reflects the test error increase shown in Fig. 2c. On the other hand, the alignment angle for augmented DFA is highly robust to the $\eta$ value and smaller than 90°. These results suggest that we can train the deep physical network using inaccurate $f'(a)$ (or even using an alternative nonlinear function), which provides ease of physical implementation.

Let us discuss the effectiveness of the augmented DFA against a randomly fixed deep network[60–62] towards application to a physical RC. Before the experiment, we investigated the applicability of DFA itself to such network using the deep RC and ELM models (see Supplementary Information S6 and S10). As in the analysis for the MLP case shown in Fig. 2a–c, we investigated robustness against the choice of $g(a)$ in the augmented DFA algorithm using the deep RC with four hidden layers. For this purpose, $f(a)$ and $g(a)$ were set to $\cos(a)$ and $\sin(a+\theta)$. The spectral radius of reservoir weight was set to 0.9 because the performance of the deep RC was maximized in this region (see Supplementary Information S13). By varying the $\theta$ value from 0 to $\pi$, we could scan $\eta$ from $-1$ to 1 easily. Figure 2d shows the test accuracy for the MNIST task as a function of $\eta$. For comparison, we also plotted the results for the same network trained by BP by replacing $f'(a)$ with $g(a)$. As can be seen, unlike for the BP training, the accuracy of the RC trained by the augmented DFA is robust against the choice of $g(a)$, which is the same trend as in the results for the MLP. The accuracy became worse when $\eta$ approached zero, and we should avoid this region to achieve better performance. These results basically support the effectiveness of the augmented DFA approach even in an RC. As shown in Supplementary Information S12, the robustness against $\eta$ highly depends on the number of nodes. This suggests that the

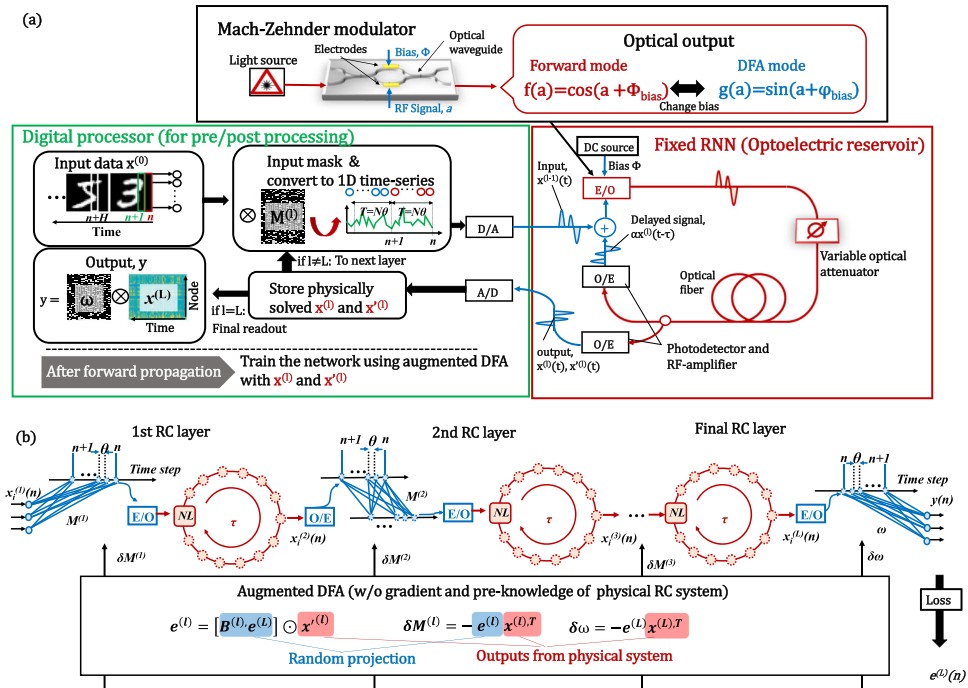

**Fig. 3 | Optoelectronic deep RC system with augmented DFA training.**
**a** Schematic of the constructed optoelectronic deep RC. The input signals are masked by a digital processor and sent to the optoelectronic RC system to solve Eqs. (4) and (5). The change in the nonlinearity from $f(a)$ to $g(a)$ is realized by applying a bias to the Mach–Zehnder modulator. Based on physically solved $x$ and $s$ values, the mask for each layer is updated. **b** Equilibrium network topology for the constructed optoelectronic RC. Each reservoir layer shows ring topology since the RC system is composed of a delay-based nonlinear fiber ring.

number of nodes is important not only for accuracy but also for robustness. The analysis of the alignment angle for this network is shown in Supplementary Information S11. In contrast to the MLP case, we found that the alignment angle for $\eta = 0$ was larger than 90°. However, even in the region beyond the alignment angle of 90°, we were able to obtain an error of around 5%. We think that the network only trains the final layer weights in this case. In our experiment, the multilayer network did not have nonlinear activation in the final layer, as in the same manner for the standard RC. As the gradient in the final layer is the same as in the standard network, the weight in the final layer is simply varied to minimize the final error even in the region with $\eta = 0$. In fact, we were able to obtain an error of approximately ~6% in the readout-only training, which supports our inference. The applications of the augmented DFA to other network models, including an MLP-Mixer, vision Transformer, and ResNet, deep-ELM are described in Supplementary Information S1 and S10.

**Physical implementation**
Here, we show a concrete example of physical implementations of PNNs trained by the augmented DFA, namely a prototype hardware/software implementation of an optoelectronic RC using an FPGA-assisted fiber-optic system. Numerical simulations of other physical implementations, including a diffractive optical neural network and a nanophotonic neural network are described in Supplementary Information S2.

Up to now, various physical implementations of single-layer RC have been achieved by using a delayed dynamical system with a single nonlinear device[8,17,18,21–23] By expanding this concept, we implemented deep RC by cascading the delayed dynamical system. Figure 3a shows a schematic explanation of our constructed optoelectronic deep RC benchtop. The equilibrium network topology is shown in Fig. 3b. In this system, temporal input signals of the $l$th layer, $x_i^{(l)}(n)$, are masked and converted to $M_i^{(l)} x_i^{(l)}(n)$ using mask function $M_i^{(l)}$, where $i$ is the virtual node number ($i = 1, 2,..., N$, where $N$ is the number of virtual nodes in each reservoir layer). This operation generates quasi-random

connections between inputs and virtual nodes. The masked inputs are converted to optical signals using an optical intensity modulator with time interval $\theta_{rc}$. Thus, the input signals are elongated to time interval $T = N\theta_{rc}$. The signals are introduced into a delay ring with a single nonlinear system, which acts as the reservoir layer. If we set the length of the delay ring $\tau$ as $\tau = T$, each node only couples with the previous state of itself [meaning $\Omega$ in Eq. (6) becomes diagonal matrix]. On the other hand, by choosing $\tau = (N + k)\theta_{rc}$, we can obtain a coupling between $x_i$ and $x_{i-k}$, which provides richer dynamics[18]. Thus, we set the delay time as the desynchronized condition $\tau = (N + 1)\theta_{rc}$. The signals are directly detected by a single photodiode, and their discretized dynamic responses are considered as virtual nodes. These signals are converted to digital signals, which are stored in the memory. Then, they are considered as the next layer input signal $x_i^{(l+1)}(n)$. They are masked by $M_i^{(l+1)}$ and re-input to the RC system. The rest of the processing scheme is the same as in the previous layer. Since this scheme shares all the hardware components, the device architecture is simple, cost-effective, and easy to implement. Other possible photonic implementations of the deep RC are summarized in Supplementary Information S3.

As a nonlinear device, we employed a Mach-Zehnder interferometer (MZM), which provides the activation $f(x) = \cos(x + \Phi_{bias})$. Then, the obtained virtual node response $x_i^{(l+1)}(n)$ can be described as

$$x_i^{(l)}(n) = \cos\left\{\alpha x_{i-1}^{(l)}(n-1) + M_i^{(l)} x_i^{(l-1)}(n) + \Phi_{bias}\right\}, \quad (4)$$

where $\alpha$ is feedback gain in the nonlinear delay ring. The operation in the next layer is the same as in the first layer. The outputs $y(n)$ are obtained from weighted summation of final layer output $x_i^{(L)}(n)$, the same as in Eq. (7) in Methods. Therefore, this stacked architecture of nonlinear delay line-based oscillators can simply emulate a special type of deep RC. For the training, we need to calculate Eqs. (8)–(11) in Methods. In particular, Eq. (9) requires heavy computational costs because it includes recurrent computation with O($N^2$) operation with each time step, the same as Eq. (6) in the forward propagation case.

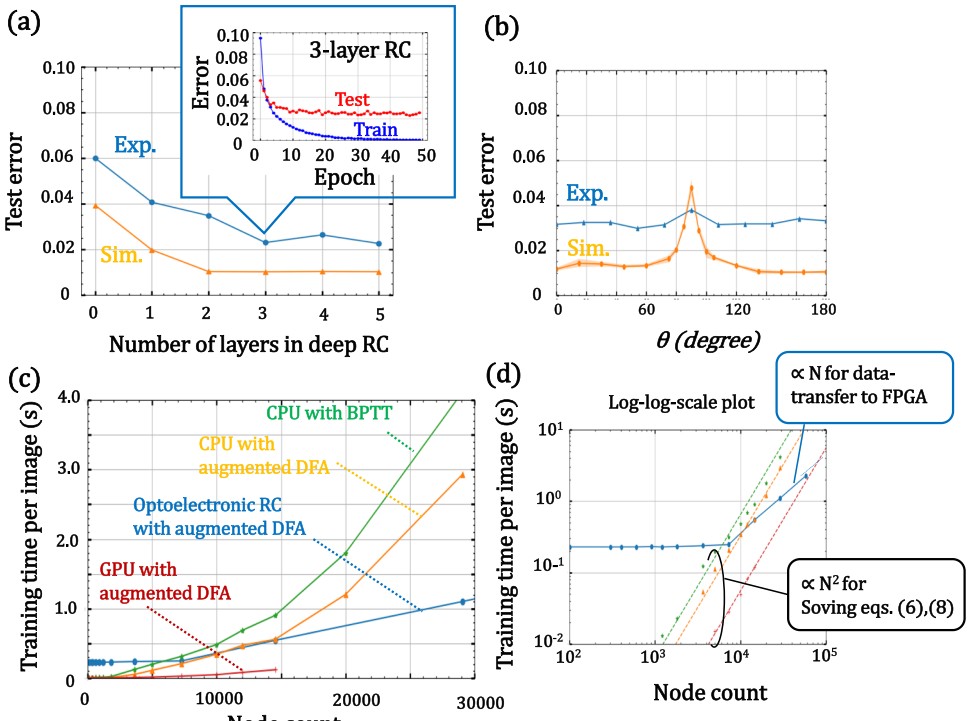

**Fig. 4 | Performance of optoelectronic deep RC system. a** Training accuracy as a function of layer number in RC with 606 nodes. Blue and orange plots show the results for the constructed benchtop and simulation on a CPU. Inset in **a** shows training and test accuracy under training. **b** Test error of the 4-layered RC with 404 nodes as a function of $\eta$. The accuracy is robust against the shape of $g(a)$.

**c** Processing time as a function of node count. The same results with a log-log-scale plot are displayed in **d**. Data in this figure were obtained using the optoelectronic RC benchtop. Reference simulation data were obtained using standard CPU and GPU computation. The test was repeated three times.

However, in this system, Eq. (9) can be solved by using the same optical system characterized by Eq. (4). By setting the bias as ($\varphi_{bias} + \pi/2$), we can generate an alternative nonlinear activation as follows:

$$s_i^{(l)}(n) = \sin\left\{\alpha s_{i-1}^{(l)}(n-1) + M_i^{(l)} x_i^{(l-1)}(n) + \varphi_{bias}\right\}. \tag{5}$$

By scanning the $\varphi_{bias}$ value, we can sweep the $\eta$ value from −1 to 1 to investigate the robustness against $\eta$. From digitally solved Eqs. (8), (10), and (11) and physically solved Eq. (9) shown in the Methods section, we can train the inherent parameters $M_i^{(l)}$ and final readout $\omega$. Note that we can solve Eq. (8) using additional optical hardware for optical random matrix operation (see Supplementary Information S9), and this approach might accelerate the computational speed further.

Based on the above proposals, we constructed a deep optoelectronic RC by combining a high-speed optoelectronic system developed for optical telecom with a highly programmable field programmable gate array (FPGA) with fast digital-to-analog/analog-to-digital converters (DAC/ADC). We also developed Pytorch/Python-compatible middleware for ease of use of our device (see Supplementary Information S7). Using the benchtop, we examined the standard benchmark tasks. Figure 4a shows the layer dependence of the test accuracy of the MNIST task (the details of the experimental setup are described in Methods; the image processing scheme using the deep RC is described in Supplementary Information S4). In this experiment, we set to the virtual node number $N = 404$, $\alpha = 0.9$, and $\varphi_{bias}$ to 0 [i.e., $g(a)$ is equal to $f(a)$ ideally]. $L = 0$ means readout-only training; $L = 1$ means both readin and readout training. As can be seen, the performance was improved by increasing the number of layers, as expected from the simulation. This suggests the effectiveness of the augmented DFA algorithm for physical RCs. Figure 4b shows the experimentally obtained test error as a function of correlation coefficient $\eta$. Note that the achieved accuracy was slightly lower than that in

Fig. 4a even for $\theta = 0$ because we stopped the training at the tenth epoch in this experiment. For comparison, the simulation results are also plotted. The robustness of the test error to the change in $g(a)$ exceeded that expected from the simulation. We think that this is due to the inherent non-idealities of the device. In our device, we changed the shape of function $g$ by applying bias voltage to optoelectronic modulator [LiNbO$_3$-based MZM (LN-MZM)]. In general, this bias voltage drifts during RC operation because the phase shift in the MZM-arm drift due to changes in temperature and humidity or some external perturbations. We compensated for such a drift effect by measuring the LN-MZM outputs every 20 minutes. However, we think that we were unable to eliminate this effect perfectly, which would explain the fluctuation of $\eta$ value in the training phase. This fluctuation smoothed out the steep dependency near the $\eta = 0$. We believe that this result is one of the useful examples of physical non-ideality for physical computing.

The obtained test accuracy for MNIST, Fashion MNIST, and CIFAR-10 are summarized in Table 1. The reported values for previous photonic DNN implementations and RC-based on other physical dynamics are also summarized in Table 1[34,35,63–69]. As references, the state-of-the-art results obtained with standard computers for these benchmarks are also shown in Table[70–72]. Despite the simplicity of our delay-based hardware implementation, we achieved performance competitively with the state-of-the-art large-scale benchtop for all the examined tasks. This supports the effectiveness of our approach.

To evaluate the efficiency of our scheme, we measured the computational time for training our system. Although previous studies also compared the processing time of PNNs, they basically only evaluated the matrix operation time [e.g., $x^{(l+1)} = f(W^{(l)} x^{(l)})$]. However, PNNs require many additional operations such as data transfer to the physical system, training based on simulation models, DAC/ADC operations, and pre- and post-processing for physical processing. Thus, the

**Table 1 | Performance comparison of PNNs for benchmark tasks**

| | | Data set | | | Training method |
|---|---|---|---|---|---|
| | | MNIST | Fashion MNIST | Cifar-10 | |
| PNN based on photonics | Deep RC [sim.] (this study) | 98.96 % (w/o preprocessing) | 86.52% | 55.80% | Augmented DFA (with optoelectronic hardware) |
| | Deep RC [exp.] (this study) | 97.80% (w/o preprocessing) | 85.91% | 47.83% | |
| | Diffractive photonic DNN | 96.6 % [exp.][34] (w/o preprocessing) | 84.6% [exp.][34] | 44.4% [exp.][69] | BP (on external standard computer) |
| | On-chip photonic DNN | 95.3% [exp.][166] (w/o preprocessing) | Not reported | Not reported | |
| | Large-scale Photonic RC | 97.15% [sim.][168] (w/o preprocessing) 98.90% [exp.][168] (with preprocessing) | Not reported | Not reported | Linear regression (on external standard computer) |
| | On-chip Photonic RC | 91.3% [exp.][35] (w/o preprocessing) | 70.1% [exp.][35] | Not reported | |
| RC based on other physical dynamics | Spintronic RC | 87.6% [sim.][65] (w/o preprocessing) | Not reported | Not reported | Linear regression (on external standard computer) |
| | Memristor RC | 88.1% [exp.][64] (w/o preprocessing) | Not reported | Not reported | |
| | Diffusive memristor RC | 83% [exp.][67] (w/o preprocessing) | Not reported | Not reported | |
| | Self-organized nanowire RC | 90.4% [exp.][63] (w/o preprocessing) | Not reported | Not reported | |
| SOTA model on standard computer (PNNs not used) | | 99.91% (Ensembled CNN)[70] | 95.99% (ResNet-110)[71] | 98.9% (Efficient Net)[72] | BP (all computations done by standard computer) |

Scores for MNIST, Fashion MNIST, and CIFAR-10 for our device. The node counts were set to 101, 202, 404, 606, 808, and the layer number was 1–5. For comparison, the reported scores for PNNs based on photonics and an RC based on other dynamics are also shown[34,35,63–69]. As references, the table shows the state-of-the-art results obtained with standard computers for these benchmarks[70–72]. The features of our approach are high performance even in a simple optical implementation and training based on optoelectronic dynamics, which can accelerate both inference and training speed. Sim. and exp. means simulation and experimental results. Preprocessing means image processing before physical inputs such as a Gabor filter[68], which can enhance performance. For a fair comparison, we focused on the results without preprocessing.

whole processing time was still under consideration. Owing to our constructed physically accelerable algorithm and its FPGA-assisted hardware implementation with full Pytorch/Python-compatible software, we can evaluate the whole processing time of our device, including the training time. As a first step, we investigated the processing time of PNNs by changing the node count, since the advantage of the physical RC approach lies in the acceleration to solve Eqs. (6) and (8) in the Methods section with O($N^2$) computational costs.

Figure 4c shows the measured training time per image for our constructed optoelectronic RC benchtop. For comparison, we also show the results for the augmented DFA and BP training on a CPU (Intel Core i7-9700, 8 cores, 3.0-GHz clock) and GPU (Nvidia Quadro P5000, 2560 cores, 16GB memory). The same results with the log-log-scaled plot are shown in Fig. 4d. The budget of processing time for the RC benchtop is broken down as follows: FPGA processing (data-transfer, memory allocation, and DAC/ADC) of ~92%; digital processing of ~8% for pre/post-processing, including the time for solving of Eqs. (7), (8), (10), and (11); and optoelectronic processing time of ~0.02% for solving Eqs. (6) and (8). Thus, the processing time is dominated by the digital computation on the FPGA and CPU in the current stage. This is because our optoelectronic benchtop implements only a reservoir layer using a single nonlinear delay line; that is, we need to transfer and measure the large-scale hidden state using a serial transmission line. These limitations can be relaxed by utilizing fully parallel and all-optical computation hardware in future[73]. As can be seen, the computation on the CPU and GPU shows O($N^2$) trends against the node count, whereas the benchtop shows O($N$), which is due to the data-transfer bottleneck. (We need O($N$) memory on the FPGA board, but the memory size on the FPGA is limited. Thus, we need to increase the number of data-transfers by reducing minibatch size, which results in a linear time increment against $N$.) The physical acceleration beyond the CPU was observed at $N$ ~5,000 and ~12,000 for the BP and augmented DFA algorithm, respectively. However, in terms of computation speed, the effectiveness against the GPU has not been directly observed yet due to the memory limitation of the GPU. By extrapolating the GPU trend, we think that we could observe physical acceleration beyond that of a GPU at N ~80,000. These estimations are on the same order as a previous estimation on the forward propagation of a photonic RC[57]. To the best of our knowledge, this is the first comparison of the whole training process and the first demonstration of physical training acceleration using PNNs.

## Discussion
### Augmentability to other physical systems

In this study, we have verified the effectiveness of our approach through physical experimentations using an optoelectronic delay-based implementation. The remaining question is its applicability to other systems. To answer it, we performed numerical simulations using a widely investigated photonic neural network, and revealed the effectiveness of our approach even in complex-valued diffractive networks and nanophotonic unitary networks (see Supplementary Information S2). In addition, our experimentally demonstrated delay-based RC was shown to be highly suitable for various physical systems. The major difference from other physical systems is the nonlinearity in Eq. (4), which is sometimes difficult to identify accurately. However, as described above, our method is highly robust to $g(a)$, which suggest the algorithm is effective for such cases. Regarding the scalability of the physical system, the major issue for constructing a deep network is its intrinsic noise. Here, we investigated the effect of noise by numerical simulation (see Supplementary Information S8 and previous works[74,75]). We found the system to be robust to noise. Regarding to scalability of RC approach to more large-scale datasets, it has been reported that an RC-based transformer model (transformer with a fixed layer trained by BP) and a vision transformer-like RC works well[76,77]. As the transformer can be applied to many practical models,

our deep RC scheme might scale to more advanced models. Further investigation will be performed in future.

## Scalability and limitation of proposed method

Current physical implementations of neural networks are mainly focusing simple models such as an RC and MLP. We demonstrated the applicability of the augmented DFA to these models through the simulations and physical experiments described above. Here, we consider the scalability of the DFA-based approach to more modern models. One of the most commonly used models for practical deep learning is a deeply connected convolutional neural network (CNN). However, it has been reported that the DFA algorithm is difficult to apply to standard CNNs[78].Thus, the proposed method may be difficult to apply to convolutional PNNs[33,67,70] in a simple manner.

On the other hand, a recent study revealed that a full-connection network named MLP-Mixer can achieve state-of-the-art performance[79]. Although DFA-based training may be effective for such convolution-free models, the applicability of DFA for MLP-Mixer has not been investigated. In addition, it has also been reported that DFA can train modern network architectures without a convolution layer, including a graph neural network and transformer[54]. Those findings suggest that our algorithm might work on such a practical network structure. Considering analog hardware implementations, the applicability to SNNs is also an important topic. The suitability of DFA-based training for SNNs has been reported[56], which implies that our proposed augmented DFA could make the training easier. Considering DFA for the CNN-based model, the investigation in a previous study was limited to the models without skip connections. It has been reported that the DFA angle increases with depth, which leads to failure of the training[78]. At the same time, it has been reported that the alignment angle in the convolution layer near the final layer is small enough even in CNNs, suggesting a shallow path to the final layer is one key to the success of the DFA-based training even in a CNN. Notably, it has been reported that forming skip connections is equivalent to forming an ensemble of deep and shallow networks. In addition, it has also been shown that most of the effective gradients in ResNet come from a shallow path. Thus, it is expected that ensembled shallow paths will have a positive impact on DFA training. In such a network, there remains the possibility of successful DFA training even for deep CNNs.

While the DFA-based algorithm has the potential to scale to above more practical models beyond a simple MLP or RC, the effectiveness of applying DFA-based training to such networks is still unknown. Here, as additional work in this research, we investigated the scalability of DFA-based training (DFA itself and the augmented DFA) to the above-mentioned models (MLP-Mixer, Vision transformer (ViT), ResNet, and SNNs). The details are described in Supplementary Information S1, and the main results for the MNIST, CIFAR-10 benchmarks for the examined results are summarized in Table 2. We found that the DFA-based training is effective even for the explored practical models. While the achievable accuracy of DFA-based training is basically lower than that of BP training, some tuning of model and/or algorithm could improve the performance. Notably, the accuracies of DFA and the augmented DFA are comparable for all the explored experimental setups, suggesting that the further improvement of the DFA itself will directly contribute to improving the augmented DFA. The results suggest that our approach is scalable to future implementation of practical models to PNNs beyond simple MLP or RC model.

## BP vs DFA in physical hardware

In general, BP is extremely difficult to implement in physical hardware because it requires all the information in the computational graph. Thus, the training of physical hardware has been done by computational simulation, which incurs large computational costs. Also, the

**Table 2 | Applicability of augmented DFA to practical network models**

| | MNIST | | | | CIFAR-10 | | | |
|---|---|---|---|---|---|---|---|---|
| | BP | DFA (Ours) | Augmented DFA (Ours) | Train only final layer | BP | DFA (Ours) | Augmented DFA (Ours) | Train only final layer |
| Custom MLP-mixer-3 | 98.90% | 98.36% | 98.49% | 92.85% | 76.50% | 62.12% | 62.01% | 43.73% |
| ResNet-18 | 99.51 | 99.41% | 99.23% | 81.70% | 91.86% | 81.68% | 80.38% | 34.92% |
| ViT-3 (w/o fine tuning) | 98.23% | 98.04% | 98.03% | 47.6% | 73.91% | 59.00% (DFA+BP) | 58.78% (DFA+BP) | 32.50% |
| SNN | 97.90% (best) 86.13% (ave.) | 96.73% (best) 93.19% (ave.) | 98.14% (best) 98.05% (ave.) | 92.21% (best) 91.95% (ave.) | — | — | — | — |

Scores for MNIST and CIFAR-10 for various ANN models trained by BP, standard DFA, and augmented DFA. As references, results for the models trained by the final layer only are shown. Accuracies for DFA and augmented DFA higher than that for final layer-only training indicate the training works effectively. Since the training became unstable with BP and the DFA algorithm when using a derivative function of spiking neurons, both the best and average scores are shown for SNN setups for fair comparison (see Supplementary Information S1.4 for the details).

difference between the model and actual system leads the degradation of accuracy. In contrast, the augmented DFA does not require accurate prior knowledge about the physical system. Thus, in deep PNNs, our DFA-based approach is more effective even in terms of accuracy than the BP-based one. In addition, the computation is accelerable by using physical hardware as demonstrated in the Results section. While our first demonstration was slower than GPU implementations, it showed the potential to accelerate the computation of both inference and training on physical hardware. In addition, the DFA training does not require sequential error propagation with layer-by-layer computation, which means that the training of each layer can be executed in parallel. Therefore, a more optimized and parallel implementation of DFA could lead to more significant speed-up. These unique features suggest the effectiveness of our DFA-based approach, especially for physical hardware-based neural networks. On the other hand, the accuracy of the model trained by the augmented DFA was still inferior to one trained by BP. Further improvement of the accuracy for DFA-based training remains future work. One approach for the improvement (combination of DFA and BP) is described in Supplementary Information S1.2.

### How to select alternative nonlinearity

In this work, we introduced alternative activation for the training. Although $g(a)$ is basically an arbitrary function, we should avoid it near $\eta = 0$. One simple way to do this is to use $g(a) = \sin(a + \theta)$. By scanning $\theta$, we can sweep the $\eta$ value for various functions and find a good solution. In addition, this nonlinearity is suitable for some physical implementations and, as shown in this article, we can accelerate the operation even in the training phase. Another approach is to use optimization problems such as a genetic algorithm (GA). Although a GA is hard to implement in a physical system, we can find a good solution for complex physical nonlinearity. An example of optimization is shown in Supplementary Information S5.

### Further physical acceleration

Our physical implementation confirmed the acceleration of recurrent processing for RC with a large-node count. However, its advantage is still limited, and further improvement is required. As mentioned in the Results section, the processing time of our current prototype is denoted as the data-transfer and memory allocation to the FPGA. Thus, integrating all the processes into the FPGA would improve the performance much more, with the sacrifice of experimental flexibility. In addition, in future, an on-board optics approach will reduce transfer cost drastically. Large-scale optical integration and on-chip integration will further improve the optical computing performance itself.

## Methods

### Augmented DFA in RC

The forward propagation of RC is given by

$$x^{(l)}(n) = f\left\{\Omega^{(l)}x^{(l)}(n-1) + M^{(l)}x^{(l-1)}(n)\right\}, \tag{6}$$

where $x^{(l)} \epsilon \mathbb{R}^{N^{(l)}}$ is the internal state of the $l$th reservoir layer $(x^{(l)}(0) = 0)$, $M^{(l)} \epsilon \mathbb{R}^{N^{(l-1)} \times N^{(l)}}$ is the connection between $(l-1)$th and $l$th reservoir layers (called a mask function), $\Omega \epsilon \mathbb{R}^{N^{(l)} \times N^{(l)}}$ is the fixed random internal connection in the $l$th reservoir layer, and $n$ is the discrete time step. The final output $y$ is obtained by

$$y(n) = \omega x^{(L)}(n) \tag{7}$$

where $\omega \epsilon \mathbb{R}^{N^{(y)} \times N^{(l)}}$ is the output weight. For the image classification task, we weighted the multiple time step signals (see Supplementary Information S4). Based on the update rule for the DFA in a recurrent neural network, gradients $\delta M^{(l)}$ and $\delta \omega$ can be calculated by using the

following equations[80].

$$e^{(l)}(n) = \left[B^{(l)}, Te^{(L)}(n)\right] \odot s^{(l)}(n), \tag{8}$$

$$s^{(l)}(n) = g(\Omega^{(l)}s^{(l)}(n-1) + M^{(l)}x^{(l-1)}(n)), \tag{9}$$

$$\delta M^{(l)}(n) = \frac{\partial E}{\partial M^{(l)}} = -e^{(l)}(n)x^{(l),T}(n), \tag{10}$$

$$\delta \omega(n) = \frac{\partial E}{\partial \omega} = -e^{(L)}(n)x^{(L),T}(n), \tag{11}$$

where $g$ is the arbitrary function, $s^{(l)} \epsilon \mathbb{R}^{N^{(l)}}$ is the auxiliary state of the $l$th reservoir layer $(s^{(l)}(0) = 0)$, and $e^{(L)}$ is the error at the final layer [see Supplementary Information S6 for the derivation and another possible candidate for Eq. (9)]. In the standard RC framework, only $\omega$ is trained by linear regression. On the other hand, our algorithm enables the training of both $\omega$ and $M^{(l)}$ for each layer. In a typical physical RC system, the operation of $M^{(l)}x(n)$ is executed by digital preprocessing. Therefore, the training $M$ is familiar with the physical implementation. Although the training of $M^{(l)}$ can be executed by BP, it requires prior knowledge of $\Omega^{(l)}$, $M^{(l)}$, and $f$. Meanwhile, augmented DFA does not require any knowledge about the physical system. By comparing with the augmented DFA for standard fully connected layers, we need to calculate Eq. (9) additionally. However, this output can be calculated by using a physical system. In addition, the DFA training does not require sequential error propagation with layer-by-layer computation, which means that the training of each layer can be executed in parallel.

### Correlation coefficient

To discuss the distance between $f'(a)$ and $g(a)$ quantitatively, we measure the correlation coefficient $\eta$ between $f'(a)$ and $g(a)$, defined as

$$\eta = \frac{\int_{-e}^{e}\left\{f'(a) - \overline{f'(a)}\right\}\{g(a) - \overline{g(a)}\}da}{\sqrt{\int_{-e}^{e}|f'(a) - \overline{f'(a)}|^2 da}\sqrt{\int_{-e}^{e}|g(a) - \overline{g(a)}|^2 da}}, \tag{12}$$

where $e$ is the natural logarithm, and the overlines mean the average. In order to discuss Eq. (12) in the bounded range where the data is distributed, the integration range is set to $[-e; e]$. The reason for this integral range is that we thought that periodic functions such as sin and cos and non-periodic functions such as tanh would yield correlations that differ from the actual situation if the range of integration is not determined. Although the distribution of internal state $a$ depends on the data set and weight value, we decided to integrate from $e$ to $-e$, assuming that the data distribution falls within this range.

### Optoelectronic benchtop

In our device shown in Fig. 3a, datasets on the standard computer were transferred to the FPGA (Xilinx Zynq, Ultra-scale) via an ethernet cable. The matrix operation of $M^{(l)}x^{(l)}$ was executed on the FPGA. Then, the signals are sent to the DAC (3-GHz, 4 GSa/s, 8-bit resolution) on the FPGA. The analog electrical signals were converted to the optical intensity by using a LiNbO$_3$-based Mach-Zehnder modulator [Thorlabs LN05FC, 32-GHz bandwidth (BW)]. After the signals had been transmitted through the optical fiber-based delay line, they were detected by a photodetector (PD) [Finisar XPRV2022, 33-GHz BW]. The detected signals were amplified by a radio-frequency (RF) amplifier [SHF-S807C (SHF), 50-GHz BW:]. The internal dynamics were received by the PD and RF amplifiers via a 1:1 splitter. The optical signal was converted to an electrical signal by the PD and then sampled by the ADC on the FPGA. The received signals were reintroduced into the optoelectronic

reservoir for the next layer calculation. After the forward propagation processing [Eq. (4)] of each minibatch, we changed the bias condition from $\Phi_{bias}$ to $\varphi_{bias}$ to change the nonlinearity from $f(a)$ to $g(a)$. Then, the same operation as the above-described forward propagation was re-executed to solve Eq. (5). After the operation, augmented DFA-based training was done on the CPU using the outputs from the optoelectronic processing. The optical system was configured to have a ring topology when the number of nodes $N = 3636$ and the sampling rate $S = 4$ GSa/s. The sampling rate can be changed under the constraint $S = S_{max}/k$, where $S_{max} = 4$ GSa/s, and $k$ is a natural number. The number of nodes can be changed by controlling $S$ under the condition $NS = constant$. The feedback gain $\alpha$ (spectral radius) can be controlled by changing the variable optical attenuator value. All the above-described processes were implemented on the Pytorch-compatible software interface described in Supplementary Information S7. Thus, we can use this optoelectronic RC like a standard CPU or GPU (in Python code, the optoelectronic device can only be described as *device="oe_rc"*). The bottleneck of the computational speed is determined by the sampling rate of the DAC/ADC. Node increments up to 29,088 displayed in Fig. 4c are realized by using the node-reuse scheme proposed by Takano et al.[81], which enables virtual node increments beyond the distance limitation of the delay ring.

## Numerical simulation

The numerical experiments were executed on a standard desktop computer with the CPU (Intel Core i7-9700, 8 cores, 3.0-GHz clock) and GPU (Nvidia Quadro P5000, 2560-core, 16GB memory). While most of the experiments were done by using our original Pytorch/Python codes, we also utilized the TinyDFA-module on the github[54] for the experiments on the commonly used ANN models (Supplementary Information S1 and Table 2). All the detailed experimental parameters are summarized in Supplementary Information S14.

## Data availability

The benchmark datasets for this work are publicly available in Pytorch or TensorFlow. All the data and methods needed to evaluate the conclusions of this work are presented in the main text and Supplementary Information. Additional data can be requested from the corresponding author.

## Code availability

Codes that are used in this paper are not available publicly due to industrial secrets. They are available from the corresponding author on reasonable request.

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

## Acknowledgements

The authors are grateful to Mr. F. Sugimoto for his support in the hardware experiment. They also thank Mr. T. Tamori for his support with the software implementation. They are also grateful to Drs. A. Samadi, T. P. Lillicrap, and D. B. Tweed for sharing the experimental codes of the SNN model. This work was partially supported by the project, JPNP16007, commissioned by the New Energy and Industrial Technology Development Organization (NEDO). K. N. was supported by JSPS KAKENHI grants numbers JP18H05472 and by JST CREST grant number JPMJCR2014.

## Author contributions

M.N., K.I., and K.N. conceived the basic concept of the presented physical deep learning method. M.N. and K.I. performed the numerical simulations. M.N. constructed the optoelectronic benchtop and executed the optical experiment. M.N. and K.T. developed the FPGA-based electric interface and Pytorch-based software implementation for the experiment. T.H., Y.K., and K.N. supervised the project. M.N. wrote the initial draft of the manuscript. All the authors discussed the results and contributed to writing the manuscript.

## Competing interests

The Authors declare no competing interests.

## Additional information

**Peer review information** : *Nature Communications* thanks Ruben Ohana and the other, anonymous, reviewers for their contribution to the peer review of this work. Peer reviewer reports are available.

