## [Peer Review File · Nature Communications]

Physical Deep Learning with Biologically Inspired Training Method: Gradient-Free Approach for Physical HardwareREVIEWER COMMENTS

Reviewer #1 (Remarks to the Author):

In “Physical Deep Learning with Biologically Plausible Training Method”, Nakajima et al. report on a concept addressing optimization of physically implemented neural networks (PNN) that takes inspiration of direct feedback alignment (DFA). DFA abolished the previous problem of error back-propagation in PNNs by removing the requirement to precisely compute the transposed matrix product and replacing it with a random matrix product. The method proposed in this manuscript goes a significant step beyond by replacing the derivative of the neurons’ nonlinear activation function essential in original DFA with an almost arbitrary function. The authors then apply their methods to a photonic PNN realization of deep reservoir computers, where intra-layer connections of the opto-electronic delay networks are implemented via an FPGA and optimized via their augmented DFA.

The manuscript makes a truly novel suggestion which could proof of great relevance to the field. Furthermore, the numerical and experimental evaluation and analysis of (a) the concept, and (b) of the experimental implementation in hardware is truly extensive. The presented methods and data in, both, numerical and experimental settings are sound. I am therefore happy to recommend publication – after some substantial improvements have been made to the manuscript’s text.

General comments:

I find the manuscript’s title misleading and not representative of the results. It is true that the authors introduce a method which could proof to be more biologically plausible. However, at no position of the manuscript the authors discuss particularities of neuro-physiological signal transduction or synaptic potentiation. Their focus clearly lies on ‘hardware plausibility’ – as it should be. In light of very high-impact publications just recently including in Nature (Wright et al.), I suggest the authors focus on that aspect and leave the proof of neuro-plausibility up to the experts of the respective field.

The manuscript would benefit significantly from some careful proof reading. The language is quite confusing at numerous places.

The same can be said for the figures. They are packed with sub-panels, which usually are not even discussed in the manuscript’s text and are therefore irrelevant for the communication to the general audience. Someone very close to the field might extract additional knowledge after very close inspection, but these details would be better allocated to the supplemental information.

More detailed comments in order of appearance:

The implementation of deep opto-electronic reservoirs were first suggested and numerically demonstrated by Penkovsky et al. PRL 2019 and should for example be included in the citations at “This has motivated proposals of deep PNNs using various physical platforms [...]”.

At least in my version the illustration of the Hadamard product operator in eq. (1) and eq. (2) fails. I assume that the superscript “T” in eq. (1) indicates the transposed matrix. In that case it should not be

in bold font.

Line 101: model[14] -> model [14]

“we can emulate 5×10^5 by 5×10^5 matrix operations on the single integrated optics [52]”. The authors cite a specific case based on random scattering, hence based on fixed weights. This should be mentioned, or the statement should be amended with a brief comments how this could be made programmable. Also, in a phase-modulator based-setting the resulting matrix would be highly constraints (not fully programmable) as it is not based on holographic imaging.

The discussion of augmented DFA, starting line 146 is not up to standard. Important information, such as network type, network size and number of layers, is not clearly provided. Do these hyper parameters remain the same for the entire discussion in this section? This part lacks clarity.

Instead of writing “cross correlation coefficient” as a figure axis, the authors could replace that with “ $\text{xcorr}(x,y)$ ” in order to unclutter Figure 2.

Line 158: “between the correlation coefficient η and ...”: correlation between what? Not clear in the text.

Line 161: how did you define outliers?

How do the authors explain the even for $\eta=0$ there is learning? Do they have an heuristic interpretation why the alignment angle still remains <90 degrees in that case? Could it be related to the linear nature of the correlation function, i.e. that a potential nonlinear functional relationship between $g(x)$ and $f(x)$ still enables convergence?

Figure 2(c): this data was obtained for which η ?

Figure 2(d) and the shown data: first you introduce shift-angles θ , and then you go back to the correlation value η . It would be helpful if you state the link between both (quite straight forward in this case). Maybe include it as a second x-axis in Fig. 2(d).

“The robustness of test error to the change in $g(x)$ exceeded that expected from the simulation. This result might originate from the noise and setting error of the physical system.” I find this, if substantiated, and outstanding yet surprising result. Could the authors elaborate a bit more on their interpretation? If this remains true than I would consider making this a more prominent result in the manuscript’s general introduction and discussion. This could be a case where hardware-inherent non-idealities actually help.

Reference [36] has now been published in Nature.

Starting line 292 the authors discuss the efficiency of their implementation, which remains limited by

the transference of matrices between hardware and FPGA. Porte, et al., Journal of Physics: Photonics 2021 have shown how this bottleneck can be removed by fully implementing all matrices in fully parallel photonic hardware.

To summarize, I find this very interesting and relevant work. If the manuscript's clarity is improved I have little doubt it will quickly play a prominent role in the field of next generation PNN hardware.

Reviewer #2 (Remarks to the Author):

Summary:

The manuscript develops an approach to deep learning for physical neural networks. The approach is based on a biologically motivated method called direct feedback alignment (DFA), which is itself based on an algorithm called feedback alignment. In essence, the family of feedback alignment (FA) algorithms compute gradient estimates for nodes deep inside the network using random feedback. The current manuscript develops an "augmented" variant of DFA that replaces the local gradient information $f'(x)$ for a layer with a random function $g(x)$. For the learning problems examined, this variant is shown to perform well in simulation as long as there is sufficient correlation between $f'(x)$ and $g(x)$. Then, the manuscript builds a proof of concept physical neural network (PNN) via an FPGA-assisted optoelectronic benchtop.

The essential idea explored in this paper — a new way of performing effective optimization for deep neural networks — is exciting and timely. The manuscript claims to provide a practical solution for training and acceleration of neuromorphic computation, along with competitive performance on relevant benchmarks. Successfully executing on this promise would be extremely useful, have many real-world consequences, and would have the potential to generate many billions of dollars in related industrial sectors (e.g. competing with existing GPUs/TPUs). Thus, the claim should be approached with the appropriate level of scrutiny. While the manuscript presents an interesting idea and set of experiments, it currently falls significantly short of a clear demonstration that the explored approach would in fact scale to real systems and datasets. So, though the manuscript is reasonably well written and may be of interest to experts in the fields of neuromorphic optimization and biological plausibility, it does not present an advance that is likely to be of interest to a wider readership.

Major Issue:

Attained performance on benchmarks is not competitive with standard approaches:

The primary benchmarking results for the algorithm (in simulation and in hardware) are presented in Table I. While these look reasonable versus some other PNN approaches, they are far from competitive

with the results expected in the field for MNIST, Fashion MNIST, CIFAR-10. For all three of these datasets, convolutional networks are standard for achieving good performance. As remarked by the manuscript, DFA has trouble optimising networks with convolutions. This was pointed out by both <https://arxiv.org/pdf/1901.01986.pdf> and earlier by <https://arxiv.org/pdf/1807.04587.pdf>. This limitation is touched briefly in the Discussion. The manuscript mentions the MLP-Mixer manuscript <https://arxiv.org/pdf/2105.01601.pdf> as reason to think that DFA-like algorithms may not in fact be required for good performance on these benchmarks. However, there may be a misunderstanding about what the MLP-Mixer manuscript demonstrates: it still makes use of localized patches (similar to a convolution), and there is every reason to believe that DFA will have trouble performing effective optimization in this case as well. The other manuscript cited to suggest that the poor performance on these benchmarks may not be an issue is: <https://arxiv.org/pdf/2006.12878.pdf>. Close reading of this manuscript shows that DFA is still significantly impaired (i.e. versus backprop) in networks with richer structure (see for example Table 5). The relatively poor performance of DFA-like algorithms in the context of standard benchmarks is unlikely to be addressed by these cited works. There may be a novel fix for this issue. This would be exciting, but would need to be demonstrated directly.

Minor Issues:

There are too many analyses of how backprop operates under impaired conditions: Figures 2a,b,d and 3c,d concern comparisons between augmented DFA and BP. This is a huge amount of figure real-estate to give to analyses whose function is unclear and does not advance the primary claim of the manuscript. Consider moving many of these to an appendix.

The augmented DFA algorithm is not significantly novel:

Prior uncited work (see [https://compneurojc.github.io/pdf/Deep Learning with Dynamic Spiking Neurons and FixedFeedback Weights.pdf](https://compneurojc.github.io/pdf/Deep%20Learning%20with%20Dynamic%20Spiking%20Neurons%20and%20FixedFeedback%20Weights.pdf)) has explored an algorithm called Broadcast Alignment, an algorithm similar to DFA. Broadcast Alignment does not extensively explore the mismatch between $f'(x)$ and $g(x)$, but it does discuss this issue and run experiments related to this possibility (see sections 2.3-2.4).

Figure 4 is difficult to understand for ML experts who do not have a hardware background.

Table 1 should include a sample of the top results for the three considered benchmarks: It would be useful to see a sample of the best scoring approaches from the field (both MLP and convolutional, and augmented versus unaugmented).

Reviewer #3 (Remarks to the Author):

In this paper, the authors develop a new method based on the Direct Feedback Alignment (DFA) training method, called augmented DFA. This method consists in replacing $f'(x)$ in the DFA update of the weights with a chosen function $g(x)$ which doesn't contain knowledge about the physical system. This allows the DFA training of Physical Neural Networks (PNNs) where information about the physical system is always complicated to obtain. The authors provide simulated and experimental results on an FPGA-assisted optoelectronic benchtop.

Overall, I think the idea of augmented DFA is a very interesting one, even if, to me it is not put at front as it should be since it is a major contribution of the paper (no explicit of augmented DFA in the abstract for instance). I think the experimental results are convincing and I expect the Pytorch/Python code to be released publicly since this can help other researchers to build on that work. The majors claims of the paper are well-supported by the experiments.

Here are the remarks/elements to improve to make the paper better:

- Important Line 303: One big flaw in this experiment is the lack of comparison with respect to a GPU. Of course you would be faster than a CPU at high dimension, but I am not sure you would be against a GPU. You should add this comparison to the paper. And it is ok if you are not faster than a GPU, since this work is a proof of concept.
- Line 38-39: the random features paper (<https://proceedings.neurips.cc/paper/2007/file/013a006f03dbc5392effeb8f18fda755-Paper.pdf>) should be cited if ELM are cited.
- Line 67: "this approach does not require error BP nor knowledge of the weight", please reformulate.
- Line 68: Cite the theoretical study of DFA (<https://arxiv.org/pdf/2011.12428.pdf>) to insist on the fact that DFA learning is supported theoretically
- Line 74: "arbitral" -> arbitrary
- Line 93: "to update W " -> W has not been defined, instead write $W^{(l)}$
- In ALL equations, the Hadamard product is not understood and is replaced by a box with a question mark, be careful.
- Equation 1, there is a comma after (1)
- Line 100: "transportation" -> transposition
- Line 112: B is a random projection, please detail the distribution you use in practice (Gaussian, Bernouilli?)
- Line 135 and in the whole paper: you say you train a Reservoir computer -> a trained RC is a Recurrent Neural Network, be specific.
- Line 158/Figure 2: please put units of the test error. Also, precise if the results are simulated or with the optoelectronic benchtop.
- Line 171: if $\eta = 0$, I don't think g is the inverse function of f , but rather its orthogonal. This claim should be double-checked.
- Figure 2: I think it would be useful to study the evolution of η as a function of the training steps. Indeed, during training, f' changes and so η . I am curious about the results.
- Line 181: the θ is not the same as the Θ of the figure, please unify this notation.

- Line 205-206: "DFA-based training is also effective in a deep neural network including passive layer": this has already been shown in (<https://arxiv.org/pdf/2101.02115.pdf> and <https://arxiv.org/pdf/2108.04217.pdf>), please cite these papers.
- Line 346: The effect of noise in DFA learning has already been studied in a differential privacy setting (<https://arxiv.org/pdf/2010.03701.pdf> and <https://proceedings.neurips.cc/paper/2021/file/b8c4c8b2271787e2f78b5fe2ce193caa-Paper.pdf>), please cite them.
- Line 415: why is your correlation function measured between $-e$ and e ? Are you sure e is the natural logarithm and not the exponential number? Please explain that better in the paper.
- Figure 2 (b): please do not plot all the seeds but instead the average with \pm the variance. This will make the figure much clearer.

If the above recommendations are taken into account, especially the important one with the GPU timing, I suggest publication of the manuscript. I think the overall idea of augmented DFA is very interesting and seems to work very well with the developed optoelectronic hardware.

Aug, 15, 2022
Nature Communications
The Macmillan Building
4 Crinan Street, London N1 9XW
United Kingdom

Dear reviewers,

We deeply thank the reviewers for taking the time to check our manuscript under the stress of COVID. We hope you and your families are well. We are also sincerely grateful for your thoughtful comments on our manuscript. We have revised the manuscript according to your advice. We have read each comment carefully, and have made corresponding corrections, which we hope address all your concerns. Before responding to each referee's comments point-by-point, we would like to first summarize the main changes we have made to the manuscript and supplementary materials.

Summary of changes:

- (1) We reconsidered the title and abstract according to R1's comment.
- (2) We added a description and numerical experiments for the implementation of the random matrix and its physical constraints (unitary DFA, intensity-only DFA) following R1's comment.
- (3) We added the numerical and experimental results for η (collation between f' and g) dependency according to R1's comment.
- (4) According to R2's comment, we added the numerical experiments for practical models (MLP-mixer, ResNet, and Vision transformer) to demonstrate the applicability of our augmented DFA and DFA itself. Please see Supplemental Material 1 for details.
- (5) We added experiments for spiking neural network and demonstrated the difference from a previous study following R2's comment. See Supplemental Material 1.4 for the details.
- (6) We added a comparison of the computation speed with a GPU in Fig. 5 according to R3's comment.
- (7) We added all the references suggested by the reviewers.
- (8) We reconstructed the figures, tables, and descriptions following the reviewer's suggestions.
- (9) We added all the experimental conditions in Supplementary Material S13 following reviewer's suggestions.
- (10) We corrected some equations because we noticed the ones in the original paper contained several mistakes.
- (11) We added some experimental data (Supplementary Material S12) and data point [Figs. 2(d), 4(b), S19(a) and S20(a)] to provide further understandings about our approaches.

To Reviewer 1

Comment 1

“Physical Deep Learning with Biologically Plausible Training Method”, Nakajima et al. report on a concept addressing optimization of physically implemented neural networks (PNN) that takes inspiration of direct feedback alignment (DFA). DFA abolished the previous problem of error back-propagation in PNNs by removing the requirement to precisely compute the transposed matrix product and replacing it with a random matrix product. The method proposed in this manuscript goes a significant step beyond by replacing the derivative of the neurons’ nonlinear activation function essential in original DFA with an almost arbitrary function. The authors then apply their methods to a photonic PNN realization of deep reservoir computers, where intra-layer connections of the opto-electronic delay networks are implemented via an FPGA and optimized via their augmented DFA.

The manuscript makes a truly novel suggestion which could proof of great relevance to the field. Furthermore, the numerical and experimental evaluation and analysis of (a) the concept, and (b) of the experimental implementation in hardware is truly extensive. The presented methods and data in, both, numerical and experimental settings are sound. I am therefore happy to recommend publication – after some substantial improvements have been made to the manuscript’s text.

We thank the referee for the positive feedback on our manuscripts. We could re-recognize the main advantage of our paper to the relevant fields thanks to your comment. To address the issues you raise below, we carefully checked your comments and revised the paper. We believe that the revisions address all your concerns.

Comment 2

I find the manuscript’s title misleading and not representative of the results. It is true that the authors introduce a method which could proof to be more biologically plausible. However, at no position of the manuscript the authors discuss particularities of neuro-physiological signal transduction or synaptic potentiation. Their focus clearly lies on ‘hardware plausibility’ – as it should be. In light of very high-impact publications just recently including in Nature (Wright et al.), I suggest the authors focus on that aspect and leave the proof of neuro-plausibility up to the experts of the respective field.

We appreciate to your suggestion about the manuscript’s title. We agree with your comment, “the authors focus on that aspect and leave the proof of neuro-plausibility up to the experts of the respective field”. Thus, we changed the title from “Physical deep learning with biologically plausible training method” to “Physical deep learning with biologically inspired training method: gradient-free approach for physical hardware”. Perhaps you may have wanted us to make more substantial changes without the word “biological” such as “Physical deep learning with augmented direct feedback alignment” or

“Physical deep neural network trained by analog-hardware-familiar training method”. In fact, we discussed whether such titles would be more suitable for our article. For the following reasons, we decided to retain the “biological” idea by using “biologically inspired training methods”. First, one of our key concepts is that the training of analog-neuromorphic-hardware should be improved by considering neurological findings, rather than by implementing current machine learning methods as is. As discussed in the main text (introduction), the training in physical NNs and biological NNs share the same incompatibility with modern machine learning algorithms. Thus, we proposed to deploy biologically inspired algorithms to physical hardware. If the title were changed to claim the development of a hardware-oriented algorithm, it would not match the explanations in the main article. Second, the word “direct feedback alignment” requires specialized knowledge in the machine learning field, which would not be suitable for broad readership of Nature communications. On the other hand, “plausible” in the original title may cause misunderstanding, as you pointed out. Also, we agree that hardware plausibility should be emphasized. Considering the comment “Overall, I think the idea of augmented DFA is a very interesting one, even if, to me it is not put at front as it should be since it is a major contribution of the paper (no explicit of augmented DFA in the abstract for instance)” from Reviewer 3, we also emphasized augmented DFA in the title using the words “Gradient-free approach for physical hardware”.

Change

We changed the title from “Physical Deep Learning with Biologically Plausible Training Method” to “Physical Deep Learning with Biologically Inspired Training Method: Gradient-Free Approach for Physical Hardware”

Comment 3

The manuscript would benefit significantly from some careful proof reading. The language is quite confusing at numerous places.

Thank you for pointing out the writing issue. We checked the manuscripts once again and rewrote some sentences. In addition, the manuscripts were also reviewed by a technical rewriting service, once again.

Changes:

We have corrected grammatical errors. Although we do not describe the point-by-point corrections here, they are highlighted in green in the revised article.

Comment 4

The same can be said for the figures. They are packed with sub-panels, which usually are not even discussed in the manuscript's text and are therefore irrelevant for the communication to the general audience. Someone very close to the field might extract additional knowledge after very close inspection, but these details would be better allocated to the supplemental information.

Thank you for your advice about the presentation issue. We have reorganized the figures to simplify them and to make them more accessible for a general audience, ~~them~~ according to comments from you and the other reviewers (e.g. R2 comment 3). Also, we added descriptions of the figures. We hope that the revisions address your concerns.

Changes:

- We moved Fig. 2(d), (f) and 3(a), (d), (e) to supplemental material.
- We added some explanations of the figures.

Comment 5

The implementation of deep opto-electronic reservoirs were first suggested and numerically demonstrated by Penkovsky et al. PRL 2019 and should for example be included in the citations at “This has motivated proposals of deep PNNs using various physical platforms [...]”.

Thank you for pointing out the relevant work. We have added the reference following your suggestion.

Changes:

- We added the suggested reference (ref. [38]).

Comment 6

At least in my version the illustration of the Hadamard product operator in eq. (1) and eq. (2) fails. I assume that the superscript “T” in eq. (1) indicates the transposed matrix. In that case it should not be in bold font. Line 101: $\text{model}[14] \rightarrow \text{model} [14]$

Thank you for pointing out the illustration errors in the equations. We used \odot and changed it to italic form. Also, we used \odot and italic form for eqs. (8)–(11).

Change

- We used \odot and italic font in eqs. (1)–(3) and (6)–(11).

Comment 7

“we can emulate 5×10^5 by 5×10^5 matrix operations on the single integrated optics [52]”. The authors cite a specific case based on random scattering, hence based on fixed weights. This should be mentioned, or the statement should be amended with a brief comment how this could be made programmable. Also, in a phase-modulator based-setting the resulting matrix would be highly constraints (not fully programmable) as it is not based on holographic imaging.

Thank you for the comment on the implementation of random matrix operation. As you pointed out, the paper [52] ([59] in the revised version) uses a fixed random scattering medium for the random matrices ($B^{(l)}$ in the main article) operation. Note that this paper used the liquid-crystal-on-silicon (LCOS) spatial-light-modulator (SLM) for the input encoding. Thus, the input vector [$e^{(L)}$ for the processing of direct feedback alignment (DFA)] is reconfigurable even in this demonstration. By using an additional SLM instead of the random scattering medium, we can implement programmable random matrices.

You also pointed out the limitation due to the phase-only modulation in the LCOS-based SLM. Before answering the question, we would like to point out that amplitude-modulation-type SLMs such as digital-mirror-devices (DMDs) can be used for the random matrix implementation, which can emulate real-valued random projection, similar to the standard DFA operation. Thus, we think it is not a critical limitation for the optical implementation. Considering the phase-only modulation, we agree with you that it may suggest that we cannot control the amplitude of weights of feedback matrices $B^{(l)}$. To address your concern, we added the simulation results for complex-valued random projection using phase-only modulated complex-valued matrices $B_{phase}^{(l)}$ for the DFA training. The details can be found in the added description in Supplemental Material S9. As can be found in the results in S9, both amplitude-only and phase-only modulation work as random matrix in the experimented condition. Thus, we think that the phase-only constrain is not critical to the DFA-based training, at least in the simple model case.

Based on the above discussion, we added a brief explanation of the device implementation in the main manuscript. Also, we added results for the DFA using random unitary matrices.

Changes:

- The optical configuration for DFA training is added briefly in the main text and in detail in the supplemental material S9.
- Comparisons with phase-only, amplitude-only, and both amplitude/phase training are added in the supplemental material S9.

Comment 8

The discussion of augmented DFA, starting line 146 is not up to standard. Important information, such

as network type, network size and number of layers, is not clearly provided. Do these hyper parameters remain the same for the entire discussion in this section? This part lacks clarity.

Thank you for pointing out the lack of the experimental information. We added the experimental conditions to the revised manuscript. Also, we summarized all experimental conditions in Supplemental Material S13.

Changes:

- We added the experimental conditions in the main text (e.g. please see p.6, L.184, 187, and 196 for Fig. 2).
- We summarized all the experimental conditions in S13.

Comment 9

Instead of writing “cross correlation coefficient” as a figure axis, the authors could replace that with “ $\text{xcorr}(x,y)$ ” in order to unclutter Figure 2.

Thank you for suggesting the shortened alternative for the figure axis.

Changes:

- We changed the x-axis of figures from “correlation coefficient between $f'(a)$ and $g(a)$ ” to “ $\text{corr}(f'(a),g(a))$ ”.

Comment 10

Line 158: “between the correlation coefficient η and ...”: correlation between what? Not clear in the text.

Thank you for pointing out the ambiguous description. We revised it as follows.

Changes:

From

Figure 2(a) shows the relationship between the correlation coefficient η and the test error after the training.

To

Figure 2(c) shows the test error as a function of correlation coefficient η (p.6, L.196, please note that Fig. 2(a) was changed to (c) in the revised version).

Comment 11

Line 161: how did you define outliers?

Thank you for the question. We defined the outliers as the values beyond the interquartile range (IQR), which is the standard way for a box-whisker plot. In our case, we used outliers to isolate the absolutely diverged data points. We added the above explanation to the manuscript.

Changes:

- Added the description of outliers (p.6, L198).

Comment 12

How do the authors explain the even for $\eta=0$ there is learning? Do they have an heuristic interpretation why the alignment angle still remains <90 degrees in that case? Could it be related to the linear nature of the correlation function, i.e. that a potential nonlinear functional relationship between $g(x)$ and $f'(x)$ still enables convergence?

Thank you for the comment. We think that the region beyond the alignment angle of 90 degrees only trains final layer weights. In our experiment, the multilayer network does not have nonlinear activation in the final layer as in the same manner for standard RC. As the gradient in the final layer is the same that in the standard network, the weight in the final layer is varied simply to minimize the final error even in the region with $\eta = 0$. In fact, we could obtain the error ~ 0.06 in the read-out-only training shown in Fig. S19(b). This value is near the error (~ 0.1) in the region with $\eta = 0$. This result supports our inference.

Change:

- We added the description about the above discussion in the main manuscript and the supplemental material S11.

Comment 13

Figure 2(c): this data was obtained for which η ?

Thank you for the comment. We obtained the data in Fig. 2(a) [Fig.2(c) in original version] from various η because the $g(x)$ was generated from $g=\tanh$, \sin , \cos , triangle , and random Fourier series and various nonlinearity. The relationship between the η and test error summarized in Fig. 2(c) [originally Fig. 2(a)]. As this might be difficult to understand, we added a description.

Change:

- We added a description to help the reader better understand the situation (p6, L187).

Comment 14

Figure 2(d) and the shown data: first you introduce shift-angles φ_{θ} , and then you go back to the correlation value φ_{η} . It would be helpful if you state the link between both (quite straight forward in this case). Maybe include it as a second x-axis in Fig. 2(d).

Thank you for the suggestion to make the figure more reader friendly. As φ_{η} and φ_{θ} have a nonlinear relationship, it is difficult to display φ_{θ} directly on the x-axis. Instead, we displayed $\cos(\varphi_{\theta})$ as the second x-axis because it has linear relationship with φ_{η} .

Change:

- We displayed $\cos(\varphi_{\theta})$ as the second x-axis in Fig. 2(d).

Comment 15

The robustness of test error to the change in $g(x)$ exceeded that expected from the simulation. This result might originate from the noise and setting error of the physical system.” I find this, if substantiated, and outstanding yet surprising result. Could the authors elaborate a bit more on their interpretation? If this remains true than I would consider making this a more prominent result in the manuscript’s general introduction and discussion. This could be a case where hardware-inherent non-idealities actually help.

Thank you for the thoughtful comment. To investigate this effect, we added data points to the Fig. 4(b) [Fig. 5(b) in the original manuscript] for both the simulation and physical experiment. As can be seen in figure, the robustness of the experiment is maintained. Considering the explanation of experimental robustness, we think that it originated from the inherent non-idealities of the device. In our device, we changed the shape of function g by applying bias voltage to optoelectronic modulator [LiNbO₃-based Mach-Zehnder modulator (LN-MZM)]. In general, this bias voltage drifts during the RC operation due to drifts of the phase shift in the MZM-arm due to temperature, humidity, or some external perturbations. We compensated for such a drift effect by measuring the LN-MZM outputs every 20 minutes. However, we think that we could not eliminate this effect perfectly, which results in the fluctuation of the η value in the training phase. This fluctuation smoothed out the steep dependency near $\eta=0$. As you pointed out, we believe that this result is one of the useful examples of physical non-

ideality for the physical computing. In the revised paper, we added the above discussion to the main manuscript.

Changes

- Data points were added in Fig.4(b)
- The discussion for the experimental robustness was added in the main text (p9, L309)

Comment 16

Reference [36] has now been published in Nature.

Thank you for bringing this to our attention. We changed the reference from arXiv version to Nature publication.

Change:

- We changed the reference from the arXiv to the Nature version (ref. [39] in the revised manuscript).

Comment 17

Starting line 292 the authors discuss the efficiency of their implementation, which remains limited by the transference of matrices between hardware and FPGA. Porte, et al., Journal of Physics: Photonics 2021 have shown how this bottleneck can be removed by fully implementing all matrices in fully parallel photonic hardware.

Thank you for providing this important information. We cited the proposed paper and added the description to reduce the transfer bottleneck.

Changes:

- Added the reference to the paper by Porte et al (ref. [74]).
- Added a description of the reduction of the digital bottleneck (p.10, L344)

Comment 18

To summarize, I find this very interesting and relevant work. If the manuscript's clarity is improved I have little doubt it will quickly play a prominent role in the field of next generation PNN hardware.

We really appreciate your thoughtful and constructive feedback. We believe that our manuscript has been highly brushed up thanks to your comments.

To Reviewer #2

Comment 1

The manuscript develops an approach to deep learning for physical neural networks. The approach is based on a biologically motivated method called direct feedback alignment (DFA), which is itself based on an algorithm called feedback alignment. In essence, the family of feedback alignment (FA) algorithms compute gradient estimates for nodes deep inside the network using random feedback. The current manuscript develops an "augmented" variant of DFA that replaces the local gradient information $f'(x)$ for a layer with a random function $g(x)$. For the learning problems examined, this variant is shown to perform well in simulation as long as there is sufficient correlation between $f'(x)$ and $g(x)$. Then, the manuscript builds a proof of concept physical neural network (PNN) via an FPGA-assisted optoelectronic benchtop.

The essential idea explored in this paper — a new way of performing effective optimization for deep neural networks — is exciting and timely. The manuscript claims to provide a practical solution for training and acceleration of neuromorphic computation, along with competitive performance on relevant benchmarks. Successfully executing on this promise would be extremely useful, have many real-world consequences, and would have the potential to generate many billions of dollars in related industrial sectors (e.g. competing with existing GPUs/TPUs). Thus, the claim should be approached with the appropriate level of scrutiny. While the manuscript presents an interesting idea and set of experiments, it currently falls significantly short of a clear demonstration that the explored approach would in fact scale to real systems and datasets. So, though the manuscript is reasonably well written and may be of interest to experts in the fields of neuromorphic optimization and biological plausibility, it does not present an advance that is likely to be of interest to a wider readership.

First, thank you for the positive comment on our essential ideas regarding to the augmented DFA and its implementation on physical neural networks. Also, we greatly appreciate the comments on the weakness of our paper, especially on the scalability. As we basically agree that your concern is an important issue for practical applications, we have added the various investigations and experiments to respond to it. The detailed results on this point can be found in our responses on your comment 2 to 6. We believe that these additional investigations could partially address your concerns.

Here, we would like to offer additional remarks on your comment, “*though the manuscript is reasonably well written and may be of interest to experts in the fields of neuromorphic optimization and biological plausibility, it does not present an advance that is likely to be of interest to a wider readership*”, because we think that the readership, even for the original version of our article, is certainly broad enough to satisfy Nature Communication requirements. Our proposal is highly relevant to the field of reservoir computing and physical implementation of neural networks. This research field is attracting the attention of many researchers in materials science and device engineering, and much

related work has been done in recent years [R1]. On the theoretical side, there are deep connections with nonlinear dynamics, chaos engineering, and information engineering, and much research has been done in these areas as well. Our proposed training method addresses key issues in these research areas, and the method is widely applicable to various physical systems as an effective candidate for a training method. We think that our claim on this point is supported by the following comment from Reviewer 1: "To summarize, I find this very interesting and relevant work. If the manuscript's clarity is improved I have little doubt it will quickly play a prominent role in the field of next generation PNN hardware".

We also understand that your comment is based the fact that the results for our physical implementation were limited to a relatively simple model (a deep RC) and simple tasks (MNIST and CIFAR10). However, let us emphasize that our demonstration is the top-level performance in the current physical implementation stage as shown in Table I. In fact, the previous demonstrations on physical hardware, even in high-impact publications (e.g. Nature-sister journals [R2-R6] and references [14-20, 33-46, 64-70] in the main article) were limited to similar (MNIST) or simpler benchmarks (e.g., XOR task) using simple network models (e.g. MLP and RC). Of course, as noted above, we agree that your concern is an important one for future practical applications. Therefore, we have performed additional experiments on the scalability of our approach, as can be found in the following responses. On the other hand, we would appreciate it if you could consider the current status of the physical implementation mentioned above when making your final decision.

[R1] You can find many related papers from the review papers on respective fields such as

(Material science) <https://www.nature.com/articles/s42254-020-0208-2>

(Photonics) <https://www.nature.com/articles/s41566-020-00754-y>

(Spintronics) <https://www.nature.com/articles/s41928-019-0360-9>

(Robotics) <https://www.nature.com/articles/s41586-021-03453-y>,

(Physical RC) <https://www.sciencedirect.com/science/article/pii/S0893608019300784>

[R2] <https://www.nature.com/articles/s41586-021-04223-6>

[R3] <https://www.nature.com/articles/s41563-021-01099-9>

[R4] <https://www.nature.com/articles/s41467-020-20719-7>

[R5] <https://www.nature.com/articles/s41586-022-04714-0>

[R6] <https://www.nature.com/articles/s41467-022-30906-3>

Comment 2

Major Issue:

Attained performance on benchmarks is not competitive with standard approaches: The primary benchmarking results for the algorithm (in simulation and in hardware) are presented in Table I. While

these look reasonable versus some other PNN approaches, they are far from competitive with the results expected in the field for MNIST, Fashion MNIST, CIFAR-10. For all three of these datasets, convolutional networks are standard for achieving good performance. As remarked by the manuscript, DFA has trouble optimizing networks with convolutions. This was pointed out by both <https://arxiv.org/pdf/1901.01986.pdf> and earlier by <https://arxiv.org/pdf/1807.04587.pdf>. This limitation is touched briefly in the Discussion. The manuscript mentions the MLP-Mixer manuscript <https://arxiv.org/pdf/2105.01601.pdf> as reason to think that DFA-like algorithms may not in fact be required for good performance on these benchmarks. However, there may be a misunderstanding about what the MLP-Mixer manuscript demonstrates: it still makes use of localized patches (similar to a convolution), and there is every reason to believe that DFA will have trouble performing effective optimization in this case as well. The other manuscript cited to suggest that the poor performance on these benchmarks may not be an issue is: <https://arxiv.org/pdf/2006.12878.pdf>. Close reading of this manuscript shows that DFA is still significantly impaired (i.e. versus backprop) in networks with richer structure (see for example Table 5). The relatively poor performance of DFA-like algorithms in the context of standard benchmarks is unlikely to be addressed by these cited works. There may be a novel fix for this issue. This would be exciting, but would need to be demonstrated directly.

Thank you for your comment about the scalability of our approach. We understood that you raised two major issues: one is the degradation of the accuracy of DFA compared with BP; the other is the applicability of DFA training to useful models such as CNNs and MLP-mixers.

On the first point, we agree that the degraded accuracy is an important issue, especially for software-level implementations. On the other hand, we would like to point out that the reviewer is making a parallel comparison between software-based and physical-hardware-based implementations. Our focus is the efficient training of physically implemented neural networks, including a physical RC (we believe there is no misrecognition on this point as you describe “the manuscript claims to provide a practical solution for training and acceleration of neuromorphic computation (from your comment 1)”). As mentioned in the main article, BP is extremely difficult to implement on physical hardware because it requires all the information in the computational graph. Thus, the training of physical hardware has been done by computational simulation, which incurs large computational costs. Also, the difference between the model and actual system leads to the degradation of accuracy. On the other hand, the augmented DFA does not require accurate pre-knowledge for a physical system. Thus, in deep PNNs, our DFA-based approach is more effective even in terms of accuracy than the BP-based one. In addition, the computation is accelerable using physical hardware as demonstrated in the results section. While our first demonstration was slower than GPU implementations, our approach has the potential to accelerate the computation of both inference and training on physical hardware as discussed in Fig. 4 (c) and (d). In addition, the DFA training does not require sequential error propagation with layer-

by-layer computation, which means that the training of each layer can be executed in parallel. Therefore, a more optimized and parallel implementation of DFA could lead to more significant speed-up. For the above reasons, we think that our DFA-based approach is more efficient than the BP-based one, especially for physical hardware applications, even if the accuracy of DFA is inferior to BP for software-level implementation. As the above points were not described well in the previous version, we have added several explanations to the discussion section in the revised manuscript.

The second concern seems to be about the problems with the DFA algorithm itself rather than the focus of our paper (nonlinearity augmentation and physical implementation). On the other hand, we also think that this point is important for the future development of our proposed approach. Therefore, we performed additional investigations to show that the application range of the DFA-based algorithm can be extended beyond the models examined in the previous version of the paper (MLP and RC). To briefly summarize, we found that the DFA-based algorithm including our approach (augmented DFA) could work well even in modern network architectures such as MLP-Mixer, Vision Transformer, and ResNet. While the achievable accuracy of DFA-based training is basically still lower than BP training, some tuning of the model and algorithm could improve the performance. Notably, the accuracies of DFA and augmented DFA are comparable to those for all the explored experimental setups, suggesting that further improvement of the DFA itself will directly contribute improving the augmented DFA. You can find the detailed descriptions of the experiments in Supplemental Material S1. In addition, we added a short summary of these investigations in the main article to address scalability.

In addition to the scalability of the DFA and augmented DFA against various models, we are now investigating the scalability of the DFA-based training towards various datasets. Here, we would like to briefly mention one obtained result of this investigation. For this research, we have introduced the GAZAN dataset [R7], one of the most widely used datasets for music genre recognition. Our numerical results (Fig. R1) show that our deep optical network successfully learns the task. The network achieved 81.6% test accuracy with BP and 79.5% with DFA for a three-second spectrogram clip, outperforming the 70% scores for humans and competitive with the 82.0% of a 12-layer ResNet setup [R8], which supports the scalability of DFA itself to real datasets. Thus, we believe that our approach is also useful for broader datasets beyond simple MNIST and CIFAR-10 datasets. Since this research is still ongoing, we would like to describe the scalability of the DFA-based training for various datasets (including Fig. R1) and its physical implementation in future publications.

Fig. R1. Layer dependence of the test accuracy for GAZAN task using (left) BP and (right) DFA training. The experimental conditions are as follows. Benchmark: GTZAN. Pre-processing: Mel spectrogram. Data augmentation: N/A. Architecture: Deep RC with various layer numbers, with feedback gain (α) of 0.9 and 1.0. Node size: 404 and 808. Optimizer: Adam with learning rate f 1.0, and nonlinearity $f(a)=\cos(a)$, $g(a)=\sin(a)$.

[R7] G. Tzanetakis et al., 2002, <https://ieeexplore.ieee.org/document/1021072>

[R8] D. Bisharad et al. 2019, <https://onlinelibrary.wiley.com/doi/abs/10.1111/exsy.12429>

Changes:

- We added a section entitled “BP vs DFA in physical network” (p12, L420)
- We fully rewrote the section “Limitation and scalability of proposed algorithm” (p11, L379).
- We added experimental results for additional models (MLP-mixer, ViT, and ResNet) trained by augmented and standard DFA, and compared them with the results for the models trained by BP (see Supplemental Material S1).
- We added a short summary of above experiments in the main manuscript and Table II.

Comment 3

There are too many analyses of how backprop operates under impaired conditions: Figures 2a,b,d and 3c,d concern comparisons between augmented DFA and BP. This is a huge amount of figure real-estate to give to analyses whose function is unclear and does not advance the primary claim of the manuscript. Consider moving many of these to an appendix.

Thank you for your comments regarding our presentation. We understand that Figs. 2(a)–(c) and 3(b), (c) are necessary parts of the presentation to show the basic functionality of a-DFA. On the other hand, the alignment analysis in Figs. 2(d)-(f) are not really necessary to understand our claim as you pointed out.

Change:

We moved Fig. 2(d), (f) and 3(a), (d), (e) from the main article to supplemental material.

Comment 4

The augmented DFA algorithm is not significantly novel:

Prior uncited work (see <https://compneurojc.github.io/pdf/Deep Learning with Dynamic Spiking Neurons and Fixed Feedback Weights.pdf>) has explored an algorithm called Broadcast Alignment, an algorithm similar to DFA. Broadcast Alignment does not extensively explore the mismatch between $f(x)$ and $g(x)$, but it does discuss this issue and run experiments related to this possibility (see sections 2.3-2.4).

Thank you for your beneficial comments and information. We have confirmed that the prior work (Samadi, A. et al., Neural computation, 2017) proposed an algorithm called broadcast alignment (BA) that is similar to the direct feedback alignment (DFA), in which the author introduced an approximated derivative function based on one tanh that does not precisely match $f'(x)$ of spiking neurons. Also, the authors mentioned in section 2.4 that the setups with the other functional forms based on logarithms are possible at the cost of convergence speed compared with the tanh-ones. The significant difference between our augmented DFA and their BA, however, lies in whether they use a priori knowledge about the system's feedforward function; that is, our augmented DFA shows that a larger variety of functions can be substituted for f' . Indeed, their BA incorporated an analytical solution of the average firing rate of the spiking neuron to obtain the approximated version of f' , and it did not explore a range of functions that greatly differ from f' . They also tried a derivative-free version, equivalent to using the identity function as g , and found that it fails to learn nearly as well as BA, by which the authors claimed that it is useful to reflect the activation function f in g for the successful training. Conversely, our additional results display that a wide range of improvised cos-based functions, which were also used in our deep optical neural networks, can obtain moderate performance competitive with tanh-ones in BA, supporting the novelty and scalability of our a-DFA compared with BA. Accordingly, we have added experimental data to the supplementary material and modified the manuscript.

Changes:

- We added experiments on the SNN using the augmented DFA approach in Supplemental Material S1.4 and Table II in the main article
- We also added a description of the suggested paper and our results in the main article (p3, L74)
- We added the suggested paper to the reference (ref. [57])

Comment 5

Figure 4 is difficult to understand for ML experts who do not have a hardware background.

Thank you for your comments. We have tried to make the figures and explanations clear. We hope we have met your expectations. We also changed Fig. 1 for the same reason.

Change:

- We reconstructed the Fig. 1 and Fig. 3 (Fig. 4 in the original version).

Comment 6

Table 1 should include a sample of the top results for the three considered benchmarks: It would be useful to see a sample of the best scoring approaches from the field (both MLP and convolutional, and augmented versus unaugmented).

Thank you for your comment. We added reported top-results in table I according to your comment. Also, we added the results for our examined architectures, including MLP (MLP-mixer), conv (ResNet) for DFA, augmented DFA, BP, and final-layer-only training in a new table II.

Finally, we really appreciate your thoughtful and constructive feedbacks. We believe that our manuscript is much improved thanks to your comments.

Changes:

- We added top results to Table I
- We added Table II

To reviewer #3

Comment 1

In this paper, the authors develop a new method based on the Direct Feedback Alignment (DFA) training method, called augmented DFA. This method consists in replacing $f'(x)$ in the DFA update of the weights with a chosen function $g(x)$ which doesn't contain knowledge about the physical system. This allows the DFA training of Physical Neural Networks (PNNs) where information about the physical system is always complicated to obtain. The authors provide simulated and experimental results on an FPGA-assisted optoelectronic benchtop.

Overall, I think the idea of augmented DFA is a very interesting one, even if, to me it is not put at front as it should be since it is a major contribution of the paper (no explicit of augmented DFA in the abstract for instance). I think the experimental results are convincing and I expect the Pytorch/Python code to be released publicly since this can help other researchers to build on that work. The majors claims of the paper are well-supported by the experiments.

We thank the referee for the positive feedback on our manuscript. Thanks to your comment, we now realize that augmented-DFA should be emphasized for the readership. Thus, we described it in the revised abstract.

To address your issues described below, we carefully checked your comments and revised the paper. We believe that the revision addresses your concerns.

Comment 2

Here are the remarks/elements to improve to make the paper better:

- Important Line 303: One big flaw in this experiment is the lack of comparison with respect to a GPU. Of course you would be faster than a CPU at high dimension, but I am not sure you would be against a GPU. You should add this comparison to the paper. And it is ok if you are not faster than a GPU, since this work is a proof of concept.

Thank you for your comment about comparison with a GPU. According to your suggestion, we added the results of GPU training in Figure 5. Unfortunately, our current implementation is slower than a GPU. However, the node count dependency of the training speed in the GPU showed an $O(N^2)$ trend. On the other hand, the training using the optoelectronic system showed an $O(N)$ trend, which suggests the potential effectiveness of the physical-system-based training. In the revised paper, we also described this point.

Changes:

- We added the GPU results in Fig. 4(c).
- We added the description for the above described GPU trends in the main article (p10, L353).

Comment 3

Line 38-39: the random features paper (<https://proceedings.neurips.cc/paper/2007/file/013a006f03dbc5392effeb8f18fda755-Paper.pdf>) should be cited if ELM are cited.

Thank you for providing the reference. We missed this pioneering paper. We cited it according to your comment.

Change:

- We added the proposed reference (ref. [26]).

Comment 4

Line 67: “this approach does not require error BP nor knowledge of the weight”, please reformulate.

Thank you for the comment. We reformulate it. In addition, we reconstructed Fig. 1 to make the features and processing of the augmented DFA clearer.

Changes:

- We revised the sentence as follows: this approach does not require layer-by-layer propagation of error signals or knowledge of the weight (p3, L. 71).
- We reconstructed Fig. 1

Comment 5

Line 68: Cite the theoretical study of DFA (<https://arxiv.org/pdf/2011.12428.pdf>) to insist on the fact that DFA

Change:

- We added the proposed reference (ref. [56]).

Comment 6

- Line 74: “arbitral” -> arbitrary
- Line 93: “to update W ” -> W has not been defined, instead write $W^{(l)}$
- In ALL equations, the Hadamard product is not understood and is replaced by a box with a question mark, be careful.

- Equation 1, there is a comma after (1)
- Line 100: “transportation” -> transposition

Thank you for pointing out the typos. We revised them in the current version of our paper.

Change:

- We revised the typos following your suggestion.

Comment 7

Line 112: B is a random projection, please detail the distribution you use in practice (Gaussian, Bernouilli?)

Thank you for pointing this out. We used a random matrix with Gaussian distribution. We added this information to the revised paper.

Change:

- We added the information about random matrix B (p. 6, L. 184).

Comment 8

Line 135 and in the whole paper: you say you train a Reservoir computer -> a trained RC is a Recurrent Neural Network, be specific.

Thank you for the comment and we apologize for the confusion. The standard RC is constructed of an input layer (fixed fully connected unit), reservoir layer (fixed recurrent unit), and output layer (trainable fully connected unit). In our paper, we only trained the input and output layer (and inter-reservoir connection in the deep RC case), and the reservoir layer is still fixed. Thus, we refer to this network as a reservoir computer (not an RNN). The reason we used such experimental set-up is follows. In the physical RC framework, only the reservoir layer is typically implemented using a physical system like in our paper. As the input layer (and inter-reservoir connection in the deep RC case) is typically implemented on a digital pre-processor, we can train these parameters to improve the performance of the physical RC in principle. However, it is difficult to apply such training with BP because the backpropagation of the physical system requires precise knowledge about the physical system and computational simulation will have large computational costs. Thus, we adopted our framework (augmented DFA) in this system. Although this point is described in the previous version of our manuscript, we realized that it is hard for the readers unfamiliar with the field to understand. Thus, we added a more detailed description about it.

Change:

- We added the description about why we refer to the experimental setup as deep RC and why we used this setup in Supplemental Material S10.

Comment 9

Line 158/Figure 2: please put units of the test error. Also, precise if the results are simulated or with the optoelectric benchtop.

Thank you for your comment. We put the units of the error according to your comment. We clarified that the experiments in the figures are simulations or using the optoelectric benchtop (please see the figure captions in the revised manuscript).

Change:

- We added the units in the Figs. 2 and 4.
- We added a description about the experimental condition in the figure captions.

Comment 10

Line 171: if $\eta = 0$, I don't think g is the inverse function of f , but rather its orthogonal. This claim should be double-checked.

Thank you for pointing out the incorrect notation. And we used the word "uncorrelation" in other part of our paper. Thus, we changed the representation "inverse function" to "uncorrelated function".

Change:

- We changed the representation from "inverse" to "uncorrelated".

Comment 11

Figure 2: I think it would be useful to study the evolution of η as a function of the training steps. Indeed, during training, f' changes and so η . I am curious about the results.

If we understand your question correctly, the η value is not changed during the training because the η is defined by the shape of functions $f'(x)$ and $g(x)$. And both the $f'(x)$ and $g(x)$ values do not vary under the training. Thus, we did not add the suggested experiment.

Comment 12

Line 181: the θ is not the same as the Θ of the figure, please unify this notation.

Very sorry for mislabeling. We revised it according to your comment.

Change:

- Θ in the figure are revised to θ . Note that the figures of concern have been moved to Supplemental Material S11 according to comments of other reviewers.

Comment 13

Line 205-206: "DFA-based training is also effective in a deep neural network including passive layer": this has already been shown in (<https://arxiv.org/pdf/2101.02115.pdf> and <https://arxiv.org/pdf/2108.04217.pdf>), please cite these papers.

Line 346: The effect of noise in DFA learning has already been studied in a differential privacy setting (<https://arxiv.org/pdf/2010.03701.pdf> and <https://proceedings.neurips.cc/paper/2021/file/b8c4c8b2271787e2f78b5fe2ce193caa-Paper.pdf>), please cite them.

Thank you for providing the important relevant papers. We cited them according to your comment. Regarding to <https://arxiv.org/pdf/2101.02115.pdf> (ref [63]), we found the formal publication [IEEE International Conference on Acoustics, Speech and Signal Processing (ICASSP) 3493–3497 (2022)]. Thus, we cited it instead of the arXiv version.

Change:

- We added the references about the DFA for passive network in the main article according to your comment (ref [62,63]).
- We also added the references about noise dependency in the supplemental material according to your comment (ref [23,24]).

Comment 14

Line 415: why is your correlation function measured between $-e$ and e ? Are you sure e is the natural logarithm and not the exponential number? Please explain that better in the paper.

Thank you for the comment. The reason for this is that we thought that periodic functions such as \sin and \cos and non-periodic functions such as \tanh would yield correlations that differ from the actual situation if the range of integration is not determined. Although the data distribution depends on the data set, we decided to integrate from e to $-e$, assuming that the distribution follows approximately a normal distribution. In the revised manuscript, we added a description about it.

Change:

- We added a description about the range of the integral (p.15 L.505).

Comment 15

Figure 2 (b): please do not plot all the seeds but instead the average with +/- the variance. This will make the figure much clearer.

Thank you for the suggestion, we changed the figures following your comment.

Change:

- We changed the figures from plotting all data to plotting the average with +/- the variance [Fig. 2(a) and (b)]

Comment 16

If the above recommendations are taken into account, especially the important one with the GPU timing, I suggest publication of the manuscript. I think the overall idea of augmented DFA is very interesting and seems to work very well with the developed optoelectronic hardware.

We really appreciate your thoughtful and constructive feedback. We believe that our manuscript has been highly brushed up thanks to your comments.

Additional changes

Correction of equations

We realized that eqs. (1)–(3) are mathematically incorrect. It was in the form for the δx , not for δW . Also, $f(x)$ should be noted as $f(a)$, where $a^{(l)} = W^{(l)}x^{(l)}$. Thus, we revised them. As the simulation and experimental setup in the original manuscript were implemented in the correct manner, these revisions do not change the results. Since we expect that it is difficult to understand how $a^{(l)}$ in the PNNs is obtained, we added a description about it in the main text.

Changes:

- We corrected the eqs. (1)–(3).
- We changed the representation of $f(x)$ and $g(x)$ to $f(a)$ and $g(a)$
- We added a description of how $a^{(l)}$ in the PNN is obtained (p.5, L.163).

Node count dependence

We added the node count dependence in S12 to provide interesting feature of augmented DFA. The results show the robustness of augmented DFA depends on the node count, suggesting that the node count is important not only for accuracy but also robustness against the alternative nonlinearity.

Change:

- We added a description of node dependency in Supplemental Material S12.

Added data points

We added the data points for the test errors and alignment analysis against θ [Figs. 2(d); simulation in 4(b); S19(a) and S20(a)] because the original plots were relatively sparse. The results show the same trend as in the original version, and the discussions is not changed.

Change:

- We added the data points [Figs. 2(d); simulation in 4(b); S19(a) and S20(a)].

Yours sincerely,
Mitsumasa Nakajima

Mitsumasa Nakajima
NTT Device Technology Laboratories
Nippon Telegraph and Telephone Corporation
3-1 Morinosato Wakamiya, Atsugi-shi, Kanagawa 243-0198, Japan
Tel: +81-46-240-2576
mitsumasa.nakajima.wc@hco.ntt.co.jp

REVIEWERS' COMMENTS

Reviewer #1 (Remarks to the Author):

The authors have very carefully addressed all of my numerous questions with detail and thought. I can recommend their manuscript for publication.

Reviewer #3 (Remarks to the Author):

I acknowledge the modifications from the authors after the rebuttal. I think this is overall an interesting paper that acts as a proof of concept, where I expect future work to build on this, in order to be closer to GPU performance.

A small note: it seems to me that the two DFA differential privacy papers (<https://arxiv.org/pdf/2010.03701.pdf> and <https://proceedings.neurips.cc/paper/2021/file/b8c4c8b2271787e2f78b5fe2ce193caa-Paper.pdf>) have still not be cited whereas they are very relevant literature of DFA + noise, please cite them.

I have also read the other reviewers comments and rebuttal from the authors, I think this paper can be accepted.

Nov, 8, 2022
Nature Communications
The Macmillan Building
4 Crinan Street, London N1 9XW
United Kingdom

Dear reviewers,

We deeply thank the reviewers for taking the time to check our manuscript under the stress of COVID. We hope you and your families are well. We are also sincerely grateful for your thoughtful comments on our manuscript. We have read each comment, and have made corresponding corrections, which we hope address your concerns. Also, we revised some points to upgrade the paper's quality. Before describing the revisions point-by-point, we would like to first summarize the main changes we have made to the manuscript and supplementary material.

Summary of changes:

- (1) We shortened the abstract to meet the editorial policy (150 words or fewer).
- (2) We deleted the conclusion section to meet the editorial policy.
- (3) We added the reference recommended by one of the reviewers.
- (4) We define $x^{(l)}(n)$ in Eq. (8) because we noticed that the definition had been deleted after the first revision.
- (5) We added the sampling points with modified simulation conditions and changed how data are displayed in Fig. 2(a)–(c).
- (6) We fixed simulation codes, and corrected the results in Fig. 4 and Table I.
- (7) We added some experimental results and descriptions in the supplementary information (Section S6 and S13)
- (8) We added the figure titles in the supplementary information to meet the editorial policy.

To Reviewer 1

Comment

The authors have very carefully addressed all of my numerous questions with detail and thought. I can recommend their manuscript for publication.

We appreciate your final positive feedback. Our manuscript has been highly improved thanks to your thoughtful and constructive comments at the first revision.

To Reviewer 3

Comment

I acknowledge the modifications from the authors after the rebuttal. I think this is overall an interesting

paper that acts as a proof of concept, where I expect future work to build on this, in order to be closer to GPU performance.

I have also read the other reviewers comments and rebuttal from the authors, I think this paper can be accepted.

We thank you for your constructive feedback. We believe that our manuscript has been highly thanks to your comments.

A small note: it seems to me that the two DFA differential privacy papers (<https://arxiv.org/pdf/2010.03701.pdf> and <https://proceedings.neurips.cc/paper/2021/file/b8c4c8b2271787e2f78b5fe2ce193caa-Paper.pdf>) have still not be cited whereas they are very relevant literature of DFA + noise, please cite them.

Thank you for pointing out the lack of the reference. The suggested papers have already been cited as references in the Supplementary Information because the analysis of noise robustness is described in section S8. In this revision, we added these references to the main manuscript.

Additional changes

Shortening of abstract

According to editorial checklists for Nature communications, we shortened the abstract (from 232 to 150 words).

Delete of conclusion

According to editorial checklists for Nature communications, we deleted the conclusion section.

Addition of experimental results for spectral radius dependency

The performance of a reservoir computer typically depends on the strength of the reservoir connection; i.e. the spectral radius of the reservoir connection, α . However, in the original article, the α dependency of the performance and the reason why we selected $\alpha=0.9$ in the experiments were unclear. Thus, we added the experimental results and descriptions in Supplementary Information S13. Also, we added the following description in the main article.

Added description in main article: The spectral radius of reservoir weight was set to 0.9 because the performance of the deep RC was maximized in this region (see Supplemental Information S13). (p.7, L.224)

Addition of training method of augmented DFA in RC

We added a description of another possible method for calculating the alternative nonlinear response for the RC model case (please see Supplementary Information S6). Interestingly, both methods (the original and added one) work well, and the achieved accuracy was almost same (please see Figs. S15 and S23 and Table SI). Although the original method is easier to implement for our optoelectronic benchtop experiment, the added method is also implementable to the physical system (please see Fig. S16), which might be useful information for some readers. We added the following description in the main article to associate the main article and supplementary information.

Added description in main article: [see Supplementary Information S6 for the derivation and another possible candidate for eq. (9)]. (p.14, L.476)

Correction of equations

We realized that the definition of $x^{(l)}(n)$ in eqs. (8) and (9) had been deleted after the first revision. We revised the description of eqs. (8) and (9) back to the original.

Correction of simulation results for Fig. 2

According to the discussion about our pre-print of this submission with researchers in the field, we modified the simulation setup of Fig. 2(a)–(c) and how data are displayed in Fig. 2(c). Regarding the latter modification, we realized that the original box plots of Fig. 2(c) might confuse readers working in a wide range of research fields because a box plot requires knowledge about how to interpret it (e.g. the meaning of the boxes and center-line correspond to the IQR and median, which are often confused with the dispersion and average). Here, we changed the plots from the original box plot [see Fig. R1(a)] to violin plots [see Fig. R1(b)]. The meaning of whiskers and plots simply correspond to the min-max and average. Also, we can see the data distribution by the filled area, as we can in a box plot. We think the modification will prevent misunderstanding of our results.

Regarding to the simulation set up, we added the sampling point for the random Fourier-series experiments [please see p 6, l.176; 50 random seeds with $f(a) = \tanh(a)$, $\cos(a)$ nonlinearity (original) revised to 100 random seeds with $f(a)=\tanh(a)$, $\cos(a)$, $\sin(a)$. Also, we changed the generation method of random Fourier-series experiments from $g(a) = r_1 + \sum_{k=1}^N p_k \sin(ka\pi) + q_k \cos(ka\pi)$ to $g(a) = \sum_{k=1}^N p_k \sin(ka\pi) + q_k \cos(ka\pi)$. The reason for erasing r_1 term is to correctly evaluate correlation coefficient η , which is defined as
$$\eta = \frac{\int_{-e}^e \{f'(a) - \bar{f}'(\bar{a})\} \{g(a) - \bar{g}(\bar{a})\} da}{\sqrt{\int_{-e}^e |f'(a) - \bar{f}'(\bar{a})|^2 dx} \sqrt{\int_{-e}^e |g(a) - \bar{g}(\bar{a})|^2 da}}$$
. In this equation, the offset term r_1 is not reflected in the difference in η . This might result in the discontinuous increment of overshooting (outlier in original Fig. R1(a); see green circle in the figure) against η value. In the modified simulation set up, we obtained the results with continuous dependency as seen in Fig.

R1(b). We think this revised simulation is more meaningful, we revised the figure. On the other hand, the obtained results even agree with the original descriptions and discussions. Thus, we only revised the description of experimental setup and explanation of displayed figure.

Fig. R1. Test error distribution of four-layer fully connected neural network as a function of the correlation coefficients between $f'(a)$ and $g(a)$, η , which is plotted by (a) box plots and (b) violin plots. Blue and red box and plots are the results for the model trained by BP and augmented DFA, respectively. η was scanned by using various $g(a)$: $\sin(a)$, $\cos(a)$, $\text{triangle}(a)$, $\text{tanh}(a)$ functions and a random Fourier series (a) with and (b) without offset term r_1 . The boxes in (a) indicate the first and third quartile of the data distribution. The whiskers in (a) mean the minimum and maximum values, excluding the outliers, and those in (b) means the minimum and maximum values of all obtained results. Filled areas in (b) mean the density of data. The region in the green circle in (a) suggests overshooting due to the offset r_1 , which was not observed in (b) due to erasing r_1 .

Changes

From: In the experiment, the MLP model was composed of four fully connected layers with 800 nodes for each layer and two types of nonlinear activation $f(a)$, namely a hyperbolic tangent (tanh) and cosine (cos) function. (p.6, L.173)

To: In the experiment, the MLP model was composed of four fully connected layers with 800 nodes for each layer and two types of nonlinear activation $f(a)$, namely a hyperbolic tangent (tanh), **sine (sin)** and cosine (cos) function.

From: Fourier series $g(a) = r_1 + \sum_{k=1}^N p_k \sin(ka\pi) + q_k \cos(ka\pi)$, where p_k , q_k , and r_1 are the random uniform coefficients sampled from $\mathbb{R} \in [-1:1]$. N was set to 4 and normalized by the relationship $|r_1| + \sum_{k=1}^N |p_k| + |q_k| = 1$. Forty random Fourier series were examined in this

experiment. Random matrix $B^{(l)}$ was generated from the uniform distribution. (p.6, L.177)

To: Fourier series $g(a) = \sum_{k=1}^N p_k \sin(ka\pi) + q_k \cos(ka\pi)$, where p_k and q_k , are the random uniform coefficients sampled from $\mathbb{R} \in [-1:1]$. N was set to 4 and normalized by the relationship $\sum_{k=1}^N |p_k| + |q_k| = 1$. A hundred random Fourier series were examined in this experiment. Random matrix $B^{(l)}$ was generated from the uniform distribution.

From: The boxes in the figure indicate the first and third quartile of the data distribution. The whiskers mean the minimum and maximum values, excluding the outliers, which are defined as the values beyond the interquartile range (IQR) to isolate the absolutely diverged data points. The outliers are also plotted as closed circles and triangles. (p.6, L.192)

To: The whiskers, plots, and filled area mean the minimum and maximum values, average values, and density of data distribution, respectively.

From: The boxes in the figure indicate the first and third quartile of the data distribution. The whiskers mean the minimum and maximum values, excluding the outliers. (figure caption for Fig.2)

To: The whiskers, plots, and filled area mean the minimum and maximum values, average values, and density of data distribution, respectively.

From: Color differences indicate differences in activation function [$f(a) = \cos(a), \tanh(a)$]. (figure caption for Fig.2)

To: Color differences indicate differences in activation function [$f(a) = \sin(a), \cos(a), \tanh(a)$].

Changes of figure

From:

To:

Correction of simulation results for Fig. 4

We noticed that the simulation codes for Fig. 4(b)–(d) and Table I were not executed in perfectly the same condition with physical implementation. Thus, we fixed the codes and re-executed the simulations to compare the results under the fair condition. As a result, the accuracies for the simulations in Table I slightly changed (from 99.03 to 98.96% for MNIST; Fashion from 86.31 to 86.52% for Fashion MNIST; from 56.85 to 55.80% for CIFAR-10). Also, the processing time for CPU and GPU simulations in Fig. 4(c) and (d) increased; e.g. the processing time for 10,000 nodes increased from 0.29 to 0.34 sec/image (CPU) and from 0.033 to 0.049 sec/image (GPU). According to these results, we revised the Fig. 4(b)–(d) and Table I. As these corrections do not significantly affect the description and discussion of the article, we only changed the description as follows

Changes

From: we could observe physical acceleration beyond that of a GPU at $N \sim 100,000$.

To: we could observe physical acceleration beyond that of a GPU at $N \sim 80,000$. (p.11, L. 355)

Changes of figure

From

To

Add figure titles in supplementary information

According to editorial checklists for Nature communications, we added the figure titles in Supplementary Information.

Add/change the description

- We added an explanation about the a-DFA to suppress the reader's misunderstanding (please see p. 6, line 204 and p.14, L.459 and L475 in the main manuscript).
- We changed "Supplementary Material" to "Supplementary Information" according to the editorial policy.
- We added some detail of experimental condition in Supplemental Information S14 (please see red characters in section S14)
- We added acknowledgement (p.22. L.715).

Yours sincerely,
Mitsumasa Nakajima

Mitsumasa Nakajima

NTT Device Technology Laboratories
Nippon Telegraph and Telephone Corporation
3-1 Morinosato Wakamiya, Atsugi-shi, Kanagawa 243-0198, Japan
Tel: +81-46-240-2576
mitsumasa.nakajima.wc@hco.ntt.co.jp